# Evolution and cell-type specificity of human-specific genes preferentially expressed in progenitors of fetal neocortex

Marta Florio[1†‡], Michael Heide[1†], Anneline Pinson[1], Holger Brandl[1], Mareike Albert[1], Sylke Winkler[1], Pauline Wimberger[2], Wieland B Huttner[1*], Michael Hiller[1,3*]

[1]Max Planck Institute of Molecular Cell Biology and Genetics, Dresden, Germany; [2]Klinik und Poliklinik für Frauenheilkunde und Geburtshilfe, Universitätsklinikum Carl Gustav Carus, Technische Universität Dresden, Dresden, Germany; [3]Max Planck Institute for the Physics of Complex Systems, Dresden, Germany

**Abstract** Understanding the molecular basis that underlies the expansion of the neocortex during primate, and notably human, evolution requires the identification of genes that are particularly active in the neural stem and progenitor cells of the developing neocortex. Here, we have used existing transcriptome datasets to carry out a comprehensive screen for protein-coding genes preferentially expressed in progenitors of fetal human neocortex. We show that 15 human-specific genes exhibit such expression, and many of them evolved distinct neural progenitor cell-type expression profiles and levels compared to their ancestral paralogs. Functional studies on one such gene, *NOTCH2NL*, demonstrate its ability to promote basal progenitor proliferation in mice. An additional 35 human genes with progenitor-enriched expression are shown to have orthologs only in primates. Our study provides a resource of genes that are promising candidates to exert specific, and novel, roles in neocortical development during primate, and notably human, evolution.
DOI: https://doi.org/10.7554/eLife.32332.001

**\*For correspondence:**
huttner@mpi-cbg.de (WBH);
hiller@mpi-cbg.de (MH)

†These authors contributed equally to this work

**Present address:** ‡Department of Genetics, Harvard Medical School, Boston, United States

**Competing interests:** The authors declare that no competing interests exist.

## Introduction

The expansion of the neocortex in the course of human evolution provides an essential basis for our cognitive abilities (*Striedter, 2005*; *Azevedo et al., 2009*; *Rakic, 2009*; *Lui et al., 2011*; *Borrell and Reillo, 2012*; *Buckner and Krienen, 2013*; *Kaas, 2013*; *Florio and Huttner, 2014*; *Dehay et al., 2015*; *Namba and Huttner, 2017*; *Sousa et al., 2017*). This expansion ultimately reflects an increase in the proliferative capacity of the neural stem and progenitor cells in the developing human neocortex (from now on collectively referred to as cortical neural progenitor cells, cNPCs) (*Azevedo et al., 2009*; *Rakic, 2009*; *Lui et al., 2011*; *Borrell and Reillo, 2012*; *Florio and Huttner, 2014*; *Bae et al., 2015*; *Dehay et al., 2015*; *Namba and Huttner, 2017*), as well as in the duration of their proliferative, neurogenic and gliogenic phases (*Lewitus et al., 2014*; *Otani et al., 2016*). It is therefore a fundamental task to elucidate the underlying molecular basis, that is, the changes in our genome that endow human cNPCs with these neocortical expansion-promoting properties.

One approach toward this goal is to identify which of the genes that are particularly active in human cNPCs exhibit a human-specific expression pattern, or even are human-specific. We previously isolated, and determined the transcriptomes of, two major cNPC types from embryonic mouse and fetal human neocortex (*Florio et al., 2015*), (i) the apical (or ventricular) radial glia (aRG), the primary neuroepithelial cell-derived apical progenitor type (*Kriegstein and Götz, 2003*; *Götz and*

*Huttner, 2005*), and (ii) the basal (or outer) radial glia (bRG), the key type of basal progenitor implicated in neocortical expansion (*Lui et al., 2011*; *Borrell and Reillo, 2012*; *Betizeau et al., 2013*; *Borrell and Götz, 2014*; *Florio and Huttner, 2014*) (*Figure 1A*). This led to the identification of 263 protein-coding human genes that are much more highly expressed in human bRG and aRG than in a neuron-enriched fraction (*Florio et al., 2015*). Of these, 207 genes have orthologs in the mouse genome but are not expressed in mouse cNPCs, whereas 56 genes lack mouse orthologs. Among the latter, the gene with the highest specificity of expression in bRG and aRG was found to be *ARHGAP11B*, a human-specific gene (*Riley et al., 2002*; *Sudmant et al., 2010*; *Antonacci et al., 2014*; *Dennis et al., 2017*) that we showed to be capable of basal progenitor amplification in embryonic mouse neocortex, which likely contributed to the evolutionary expansion of the human neocortex (*Florio et al., 2015*; *Florio et al., 2016*).

Our previous finding that, in addition to *ARHGAP11B*, 55 other human genes without mouse orthologs are predominantly expressed in bRG and aRG (*Florio et al., 2015*), raises the possibility that some of these genes also may be human-specific and may affect the behaviour of human cNPCs. To investigate the evolution and cell-type specificity of expression of such genes, we have now data-mined our previous dataset (*Florio et al., 2015*) as well as four additional ones (*Fietz et al., 2012*; *Miller et al., 2014*; *Johnson et al., 2015*; *Pollen et al., 2015*) to carry out a comprehensive screen for protein-coding genes preferentially expressed in cNPCs of fetal human neocortex. We find that, in addition to *ARHGAP11B*, 14 other human-specific genes show preferential expression in cNPCs. Furthermore, we identify 35 additional human genes exhibiting such expression for which orthologs are found in primate but not in non-primate mammalian genomes. We provide information on the evolutionary mechanisms leading to the origin of several of these primate-specific genes, including gene duplication and transposition. Moreover, we analyze the cell-type expression patterns of most of the human-specific genes, including expression of their splice variants. By comparing the expression of the human-specific genes with their respective ancestral paralog, we show a substantial degree of gene expression divergence upon gene duplication. Finally, we show that expressing the human-specific cNPC-enriched *NOTCH2NL* gene in embryonic mouse neocortex promotes basal progenitor proliferation. Our study thus provides a resource of genes that are candidates to exert specific roles in the development and evolution of the primate, and notably human, neocortex.

## Results

### Screen of distinct transcriptome datasets from fetal human neocortex for protein-coding genes preferentially expressed in neural stem and progenitor cells

To identify genes preferentially expressed in the cNPCs of the fetal human neocortex, we analyzed five distinct, published transcriptome datasets obtained from human neocortical tissue ranging from 13 to 21 weeks post-conception (wpc). First, the RNA-Seq data obtained from specific neocortical zones isolated by laser capture microdissection (LCM) (*Fietz et al., 2012*), which we screened for all protein-coding genes that are more highly expressed in the VZ, iSVZ and/or oSVZ than the cortical plate (CP) (*Figure 1A,B*). This yielded 2780 genes (*Figure 1D*). Second, the Allen Brain Institute microarray data (BrainSpan Atlas) obtained from LCM-isolated specific neocortical zones (*Miller et al., 2014*) (*Figure 1A,B*), which we screened for all protein-coding genes with positive laminar correlation with either the VZ, iSVZ or oSVZ as compared to the zones enriched in postmitotic cells (intermediate zone (IZ), subplate, CP, marginal zone, subpial granular zone). This yielded 3802 genes (*Figure 1D*). Third, the RNA-Seq data obtained from specific neocortical cell types isolated by fluorescence-activated cell sorting (FACS) (*Florio et al., 2015*), which we screened for all protein-coding genes more highly expressed in aRG and/or bRG in S-G2-M as compared to the cell population enriched in postmitotic neurons but also containing bRG in G1 (*Figure 1A,B*). This yielded 2030 genes (*Figure 1D*). Fourth, the data obtained from single-cell RNA-Seq of dissociated cells captured from microdissected VZ and SVZ (*Pollen et al., 2015*), which we screened for all protein-coding genes positively correlated with either radial glial cells, bIPs or both (*Figure 1A,B*) and negatively correlated with neurons. This yielded 4391 genes (*Figure 1D*). Fifth, the transcriptome-wide RNA-Seq data obtained from specific neocortical cell types isolated by FACS (*Johnson et al., 2015*),

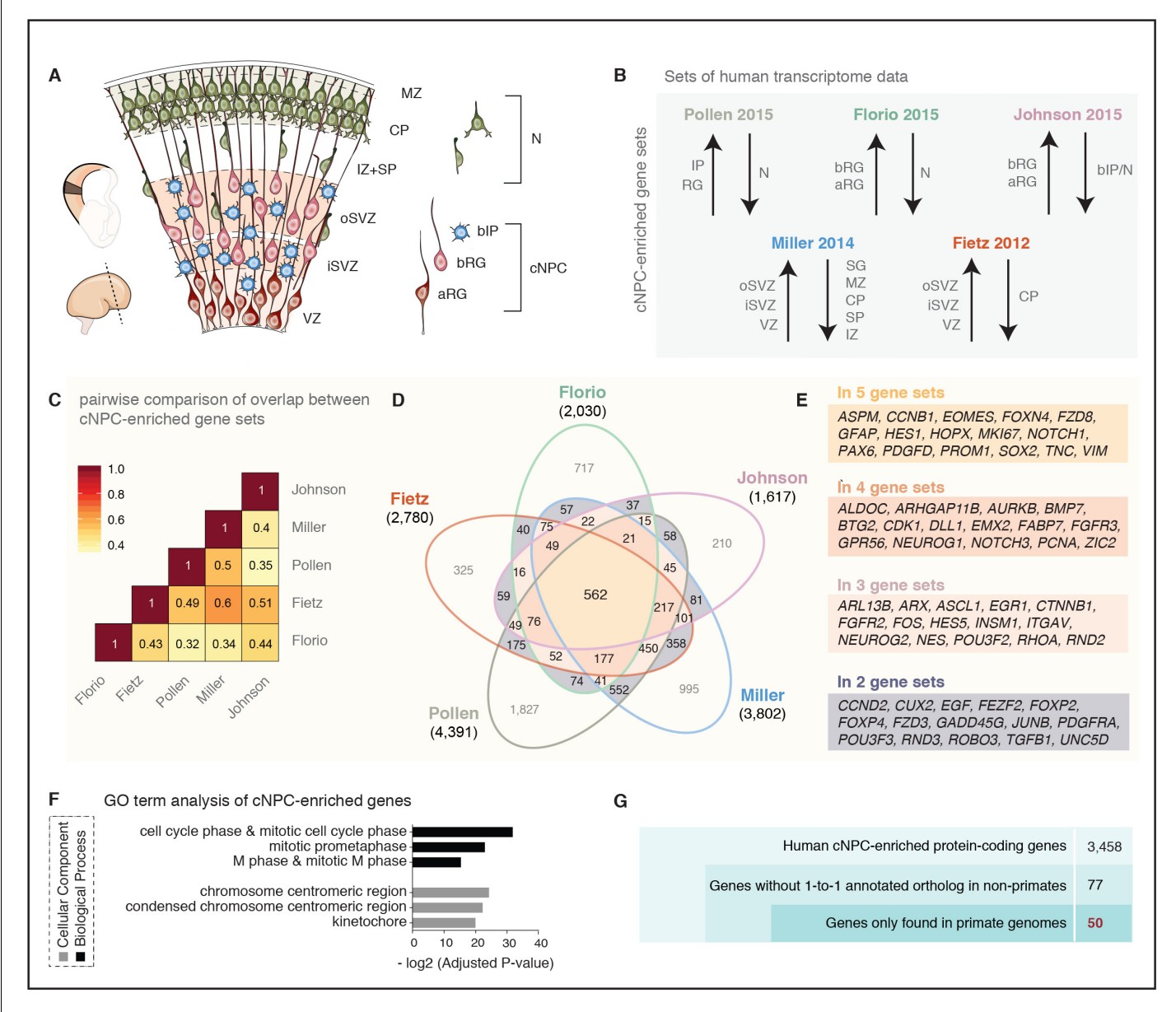

**Figure 1.** A screen for human cNPC-enriched protein-coding genes and determination which of them have orthologs only in primates. (**A**) Cartoon illustrating the main zones and neural cell types in the fetal human cortical wall that were screened for differential gene expression in the human transcriptome datasets as depicted in (**B**). Adapted from (*Florio et al., 2017*). SP, subplate; MZ, marginal zone. (**B**) The indicated five published transcriptome datasets from fetal human neocortical tissue (*Fietz et al., 2012*; *Miller et al., 2014*) and cell populations (*Florio et al., 2015*; *Johnson et al., 2015*; *Pollen et al., 2015*), were screened for protein-coding genes showing higher levels of mRNA expression in the indicated germinal zones and cNPC types than in the non-proliferative zones and neurons. (**C**) Heat map showing a pairwise comparison of the degree of overlap between the five gene sets of human genes with preferential expression in cNPCs. (**D**) Venn diagram showing the gene sets of human protein-coding genes displaying the differential gene expression pattern depicted in (**B**). Numbers within the diagram indicate genes found in two (violet), three (pink), four (orange) or all five (yellow) gene sets. Genes found in at least two gene sets were considered as being cNPC-enriched. (**E**) Selected genes with established biological roles found in two, three, four, or all five gene sets. (**F**) GO term analysis of human cNPC-enriched genes. The top three most enriched terms for the category *Cellular Component* (black bars) and for the category *Biological Process* (grey bars) are shown. (**G**) Stepwise analysis leading from the 3458 human cNPC-enriched protein-coding genes to the identification of 50 primate-specific genes.

DOI: https://doi.org/10.7554/eLife.32332.002

The following figure supplement is available for figure 1:

**Figure supplement 1.** Occurrence of the 50 primate-specific genes in the five gene sets.

DOI: https://doi.org/10.7554/eLife.32332.003

which we screened for all protein-coding genes more highly expressed in aRG and/or bRG as compared to the cell population enriched in bIPs and neurons (*Figure 1A,B*). This yielded 1617 genes (*Figure 1D*).

Since these transcriptome datasets were obtained using different cell identification and isolation strategies (i.e. LCM, FACS, single-cell capture), gestational ages (13–21 wpc), and sequencing technologies (i.e. RNA-Seq, microarray, single-cell RNA-seq) (see *Supplementary file 1* for details), we investigated to what extent the sets of genes preferentially expressed in cNPCs overlapped across datasets (*Figure 1B,C*). Pairwise comparison of these cNPC-enriched gene sets revealed a substantial overlap between datasets, ranging from 32% (Florio-Pollen) to 60% (Fietz-Miller) (*Figure 1C*). Thus, despite the differences in the experimental approaches used to generate the original datasets, a substantial proportion of the protein-coding genes here identified using the above-described criteria were the same.

Next, we determined how many of the protein-coding genes exhibiting the above-described differential expression pattern were found in all five gene sets. This was the case for 562 genes (*Figure 1D*, yellow). We also determined the number of genes found in four of the five gene sets (five combinations, *Figure 1D*, orange), in three of the five gene sets (10 combinations, *Figure 1D*, pink) and in two of the five gene sets (10 combinations, *Figure 1D*, violet). Together this yielded a catalogue of 3458 human genes with preferential expression in cNPCs in at least two gene sets (from here on referred to as cNPC-enriched genes) (see *Supplementary file 1*).

These 3458 genes included many canonical markers of cNPCs and known molecular players involved in (i) function of radial glia (e.g. *FABP7, GFAP, NES, VIM, PROM* (*Taverna et al., 2014*), *PAX6* (*Osumi et al., 2008*; *Walcher et al., 2013*), *SOX2* (*Lui et al., 2011*), *HOPX* (*Pollen et al., 2015*)) and intermediate progenitors (e.g. *EOMES* (*Englund et al., 2005*), *NEUROG1/2* (*Fode et al., 2000*; *Schuurmans et al., 2004*)), (ii) cell proliferation (e.g. *MKI67, PCNA*), (iii) Notch signaling (e.g. *DLL1, HES1, HES5, NOTCH1, NOTCH3* (*Kageyama et al., 2009*; *Imayoshi et al., 2013*)), and (iv) extracellular matrix and growth factor signaling (e.g. *EGF, FGFR2, FGFR3* (*Vaccarino et al., 1999*), *ITGAV* (*Stenzel et al., 2014*), *TGFB1, TNC* (*von Holst et al., 2007*)) (listed in *Figure 1E*). Moreover, several genes recently implicated in human-specific aspects of cNPC proliferation and neocortex formation (*Geschwind and Rakic, 2013*; *Lui et al., 2014*; *Bae et al., 2015*; *Florio et al., 2017*; *Heide et al., 2017*; *Mitchell and Silver, 2018*; *Sousa et al., 2017*) (e.g. *ARHGAP11B, FOXP2, FZD8, GPR56, PDGFD*) were found in the analyzed gene sets, although not necessarily in all five (*Figure 1E*).

We validated the set of 3458 cNPC-enriched genes by performing a gene ontology (GO) term enrichment analysis (see *Figure 1D* and *Supplementary file 1*). This revealed that for the category *Biological Process* the GO terms 'cell cycle phase and mitotic cell cycle phase', 'mitotic prometaphase' and 'M phase and mitotic M phase' were the three most enriched ones, and for the category *Cellular Component* the GO terms 'chromosome centromeric region', 'condensed chromosome centromeric region' and 'kinetochore' were the three most enriched ones (*Figure 1F*, *Supplementary file 2*). This underscored that the cNPC-enriched genes identified here preferentially encode proteins involved in cell division, including core components of the mitotic machinery.

The catalog of the 3458 cNPC-enriched human genes presented here (*Supplementary file 1*) provides a resource (i) to interrogate the cNPC enrichment of candidate genes of interest and (ii) to potentially uncover new genes involved in cNPC function during fetal human corticogenesis.

## Identification of primate-specific genes

Primate-specific, notably human-specific, genes expressed in cNPCs have gained increasing attention for their potential role in species-specific aspects of neocortical development, including neurogenesis (*Charrier et al., 2012*; *Dennis et al., 2017*; *Florio et al., 2017*; *Heide et al., 2017*; *Sousa et al., 2017*). To determine how many of the 3458 human cNPC-enriched protein-coding genes have orthologs only in primates but not in non-primate species, we excluded from this gene set all those genes with an annotated one-to-one ortholog in any of the sequenced non-primate genomes (*Figure 1G*). This greatly reduced the number of genes from 3458 to 77 genes.

Next, we examined these 77 genes to extract those that are truly primate-specific. By inspecting genomic alignments, gene neighborhoods and gene annotations in primate and non-primate mammals, we concluded that 27 of these genes likely have an ortholog in non-primate mammals and we therefore excluded them from further analysis. The remaining 50 genes were considered to be truly

primate-specific (*Figure 1G*, *Figure 1—figure supplement 1*) and are of special interest as they may have contributed to neocortical expansion during primate evolution.

## Phylogenetic analysis of the primate-specific genes

To trace the evolution of the 50 primate-specific genes and to infer their ancestry, we investigated in which species these genes exhibit an intact reading frame and used this information to assign each gene to a primate clade. First, we found that 25 of these 50 genes predate the ape (Hominoidea) ancestor and that 14 of these 25 genes encode zinc finger proteins (*Figure 2A*, *Table 1* and *Supplementary file 3*). Remarkably, 15 of the remaining 25 genes that postdate the ape ancestor are only present in the human genome, and thus arose (or evolved to their present state) in the human lineage after its split from the lineage leading to the chimpanzee (*Figure 2A*, *Table 1* and *Supplementary file 3*) (~5–7 Mya, (*Brunet et al., 2002*; *Vignaud et al., 2002*; *Brunet et al., 2005*)). These 15 human-specific genes include *ARHGAP11B*, a gene that we reported previously to promote cNPC proliferation and neocortex expansion (*Florio et al., 2015*; *Florio et al., 2016*) and that was also present in the archaic genomes of Neandertals and Denisovans (*Sudmant et al., 2010*; *Meyer et al., 2012*; *Antonacci et al., 2014*; *Prüfer et al., 2014*; *Florio et al., 2015*).

Similar to *ARHGAP11B*, 13 of the remaining 14 human-specific genes existed also in the genomes of Neandertals and Denisovans (*Sudmant et al., 2010*; *Dennis et al., 2017*) and present data (see *Table 1*) and thus arose before the split of the lineages leading to modern humans vs. Neandertals/Denisovans ~500,000 years ago (*Meyer et al., 2012*; *Prüfer et al., 2014*). The only one of the 15 human-specific genes that has been reported to have arisen in the lineage leading to modern humans after its divergence from the lineage leading to Neandertals and Denisovans is *SMN2* (*Dennis et al., 2017*). The *SMN2* gene can alleviate spinal muscular atrophy, a neurological disease caused by mutations of *SMN1* (the ancestral paralog of *SMN2*) (*Parsons et al., 1996*; *Lorson et al., 1999*; *Watihayati et al., 2009*), and thus can be regarded as an, albeit inefficient, *SMN1* back-up specific to modern humans.

Next, we asked whether the rate at which new cNPC-enriched genes arose during primate evolution was relatively constant, or whether there were perhaps bursts in the appearance of new cNPC-enriched genes at certain steps during primate evolution. To address this question, we plotted the number of new cNPC-enriched genes that appeared in the various primate clades as a function of the length of the respective branch (see *Figure 2A*) (measured as the rate of neutral base pair substitutions). This revealed a disproportionately high rate of appearance of new cNPC-enriched genes in two of the branches, the branch that leads to Catarrhini (branch 6) and the branch that leads to human (branch 1) (*Figure 2B*).

## Analysis of selected primate-specific genes reveals distinct evolutionary mechanisms

Next, we examined how these 50 primate-specific genes evolved. We first focused on three primate-specific genes that are not human-specific, which we selected in light of their potential biological role – *MICA*, *KIF4B* and *PTTG2*.

*MICA* (*MHC class I polypeptide-related sequence A*) is the only gene among the 50 primate-specific genes analyzed in the present study that has an established relationship to the MHC locus (*Bahram et al., 1994*), pointing to a possible primate-specific interaction between cNPCs and cells of the immune system. *MICA* is a paradigmatic example of a gene arising by gene duplication (*Bailey et al., 2002*; *Eichler et al., 2004*; *Fortna et al., 2004*; *Hurles, 2004*), a well-known driving force of genome evolution (*Lynch and Conery, 2000*). *MICA* arose by duplication of the *MICB* gene, and this event occurred in the Catarrhini ancestor (*Figure 2A*).

Besides gene duplication, however, other mechanisms were found to contribute to the evolution of primate-specific genes. A notable example is *KIF4B* (*Kinesin Family Member 4B*), a gene encoding a kinesin involved in spindle organization during cytokinesis (*Zhu et al., 2005*). In fact, *KIF4B* is the only member of the kinesin superfamily among the 50 primate-specific genes. *KIF4B* evolved in the Simiiformes ancestor (*Figure 2A*) by retroposition of *KIF4A*, a gene with a near-ubiquitous occurrence in the animal kingdom (*Hirokawa et al., 2009*). This retroposition involved the reverse transcription of a spliced *KIF4A* mRNA followed by insertion of the DNA into the genome as an intronless copy of *KIF4A*.

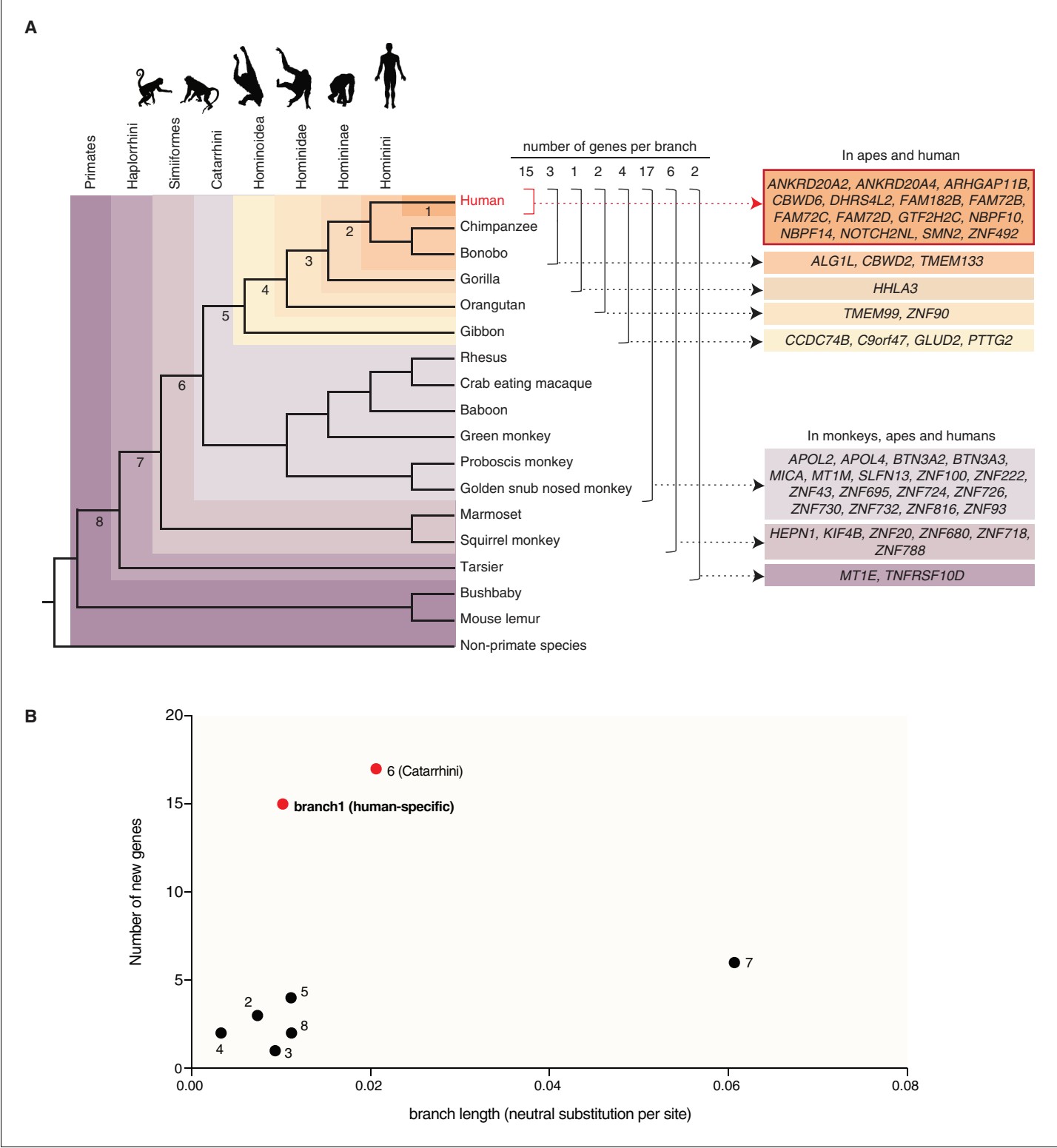

**Figure 2.** Occurrence of the primate-specific genes in the various primate clades. (**A**) Assignment of the 50 primate-specific genes to a primate clade, based on the primate genome(s) in which an intact reading frame was found in the present analysis. Clades are specified on the top left. The color-coding and brackets indicate the species in each clade analyzed in the present study. Numbers on top of the brackets indicate the number of genes assigned to that clade. Note that the occurrence of the genes in the various clades does not necessarily apply to every species in the clade. (**B**) Diagram depicting the number of new cNPC-enriched genes as a function of the frequency of occurrence of neutral base pair substitutions in the eight

*Figure 2 continued on next page*

Figure 2 continued
different branches leading to these various clades (branch length). Numbered dots indicate the branches shown in panel (A). Red dots indicate the branches with disproportionately high rates of appearance of new cNPC-enriched genes.

DOI: https://doi.org/10.7554/eLife.32332.004

The third gene we analyzed was *PTTG2* (*pituitary tumor transforming 2*), as its paralog *PTTG1* encodes a tumorigenic protein implicated in promoting proliferation of pituitary tumor cells (*Domínguez et al., 1998*; *Zhang et al., 1999*; *Vlotides et al., 2007*). Similar to *KIF4B*, the primate-specific gene *PTTG2* arose by retroposition of a reverse transcribed spliced mRNA of *PTTG1*, a gene encompassing five protein-coding exons that are conserved in reptiles, birds and mammals. However, while *KIF4B* inserted into an intergenic locus (that however allowed its transcription), the intronless protein-coding *PTTG2* inserted in sense direction into intron 2 of the *TBC1D1* gene (*Figure 3A*), which encodes a Rab-GTPase activating protein (*Roach et al., 2007*). Remarkably, whereas the *PTTG2* retroposition event already occurred in the Simiiform ancestor, the *PTTG2* gene underwent two principally different lines of evolution after retroposition. In all non-Hominoidea Simii-formes (see *Figure 2A*), consistent with neutral evolution, *PTTG2* accumulated frameshifting dele-tions and translational stop codon mutations that cause a premature termination of the open-reading frame (*Figure 3B*). In contrast, in Hominoidea (apes and humans, see *Figure 2A*), the *PTTG2* reading frame remained open, with one noticeable change. This is a 1 bp insertion (T, see *Figure 3A*) near the 3' end of the *PTTG2* open-reading frame that causes a shift in the reading frame, resulting in a new 13-amino acid-long C-terminal sequence of PTTG2 in great apes (including human) (as opposed to 24 amino acids in PTTG1) (*Figure 3B*). This PTTG2-specific sequence lacks the cluster of acidic residues found in the C-terminal sequence of PTTG1. In the case of the gibbon, however, the *PTTG2* gene carries (in addition to the 1 bp T insertion) a 22 bp deletion a few nucleo-tides 5' to this insertion. This causes yet another shift in the reading frame that results in the replace-ment of the C-terminal 25 amino acids of the PTTG2 of great apes (including human) by an 18-amino acid-long sequence (*Figure 3B*). The potential consequences of these changes in protein sequence for the function of PTTG2 with regard to cell proliferation are discussed below.

## Evolutionary mechanisms that gave rise to the human-specific cNPC-enriched protein-coding genes

We next investigated how the 15 human-specific cNPC-enriched protein-coding genes evolved. Twelve of them arose by duplications of entire genes (*Bailey et al., 2002*; *Eichler et al., 2004*; *Fortna et al., 2004*; *Hurles, 2004*) (*Figure 4A*). A special case, illustrating possible evolutionary tra-jectories after gene duplication, is the human-specific cNPC-enriched *NOTCH2NL* gene (*Figure 4A*). It arose from a duplication that included the genes *NBPF7*, *ADAM30*, and *NOTCH2* (*Figure 4—fig-ure supplement 1*). After the gene duplication event, a deletion occurred that removed the dupli-cated *ADAM30* gene and a large portion of the duplicated *NOTCH2* gene. This portion included most of the sequence giving rise to the two long *NOTCH2* splice variants (ENST00000256646 and ENST00000579475). The remaining duplicated *NOTCH2* gene sequence (the *NOTCH2NL* gene) gives rise to a short splice variant (ENST00000602566) (*Figure 4—figure supplement 1*) that enco-des only a short segment of the NOTCH2 ectodomain.

One gene, *ARHGAP11B*, arose by a partial gene duplication (*Figure 4B*). Its evolution has been analyzed previously (*Riley et al., 2002*; *Antonacci et al., 2014*; *Florio et al., 2015*; *Florio et al., 2016*; *Dennis et al., 2017*; *Dougherty et al., 2017*).

The remaining two of the 15 human-specific cNPC-enriched protein-coding genes evolved in dis-tinct ways. The *ZNF492* gene as such exists in the genomes of all non-human great apes. In the case of human, however, an exon of another zinc finger protein-encoding gene, *ZNF98*, inserted into the *ZNF492* locus, yielding a chimeric human-specific protein containing the repressor domain of ZNF492 and the DNA binding domain of ZNF98 (*Figure 4C*). The *FAM182B* gene as such exists not only in human but also in chimpanzee, bonobo and gorilla. However, in bonobo and gorilla, a stop codon terminates the potential open-reading frame soon after the initiator methionine, whereas in human a single T→G substitution replaces this premature stop with a sense codon to yield a 152-amino acid-long protein (*Figure 4D*). In chimpanzee, the T of the TAG stop codon is deleted, which

**Table 1.** Primate-specific genes

| Gene symbol | Gene name | Function | cNPC-enriched in | Occurrence | Features |
|---|---|---|---|---|---|
| *ANKRD20A2* | Ankyrin repeat domain 20 family member A2 | Unknown | Florio, Pollen, Miller | Homo (before Neandertal-Denisovan split) | Five ankyrin repeats, three coiled coil motifs [UniProt] |
| *ANKRD20A4* | Ankyrin repeat domain 20 family member A4 | Unknown | Florio, Fietz, Pollen | Homo (before Neandertal-Denisovan split) | Five ankyrin repeats, three coiled coil motifs [UniProt] |
| *ARHGAP11B* | Rho GTPase activating protein 11B | Basal progenitor amplification (***Florio et al., 2015***) | Florio, Fietz, Pollen | Homo (before Neandertal-Denisovan split) | One nucleotide substitution led to a novel splice donor site in exon five resulting in a novel and unique C-terminal sequence and a loss of Rho-GAP activity (***Florio et al., 2015***; ***Florio et al., 2016***) |
| *CBWD6* | COBW Domain Containing 6 | Unknown | Pollen, Miller | Homo (before Neandertal-Denisovan split) | CobW domain, ATP binding sites [UniProt] |
| *DHRS4L2* | Dehydrogenase/reductase 4 like 2 | Maybe an NADPH dependent retinol oxidoreductase [RefSeq] | Fietz, Pollen | Homo (before Neandertal-Denisovan split) | Unknown |
| *FAM182B* | Family with sequence similarity 182 member B | Unknown | Fietz, Miller | Homo (before Neandertal-Denisovan split) | Removal of a stop codon resulting in an open reading frame in humans (this publication) |
| *FAM72B* | Family with sequence similarity 72 member B | Unknown | Florio, Fietz, Pollen | Homo (before Neandertal-Denisovan split) | Unknown |
| *FAM72C* | Family with sequence similarity 72 member C | Unknown | Florio, Fietz, Pollen | Homo (before Neandertal-Denisovan split) | Unknown |
| *FAM72D* | Family with sequence similarity 72 member D | Unknown | Florio, Fietz, Miller | Homo (before Neandertal-Denisovan split) | Unknown |
| *GTF2H2C* | GTF2H2 family member C | Unknown | Pollen, Miller | Homo (before Neandertal-Denisovan split) | VWFA domain, C4-type zinc finger motif [UniProt] |
| *NBPF10* | Neuroblastoma Breakpoint Family Member 10 | Contains DUF1220 domains which have been implicated in a number of developmental and neurogenetic diseases (e.g. microcephaly, macrocephaly, autism, schizophrenia, cognitive disability, congenital heart disease, neuroblastoma, and congenital kidney and urinary tract anomalies) [RefSeq] | Fietz, Pollen | Homo (before Neandertal-Denisovan split) | Tandemly repeated copies of DUF1220 protein domains [RefSeq], coiled coil domain [UniProt] |

*Table 1 continued on next page*

*Table 1 continued*

| Gene symbol | Gene name | Function | cNPC-enriched in | Occurrence | Features |
|---|---|---|---|---|---|
| NBPF14 | Neuroblastoma Breakpoint Family Member 14 | Contains DUF1220 domains which have been implicated in a number of developmental and neurogenetic diseases (e.g. microcephaly, macrocephaly, autism, schizophrenia, cognitive disability, congenital heart disease, neuroblastoma, and congenital kidney and urinary tract anomalies) [RefSeq] | Fietz, Pollen | Homo (before Neandertal-Denisovan split) | Tandemly repeated copies of DUF1220 protein domains [RefSeq], coiled coil domain [UniProt] |
| NOTCH2NL | Notch 2 N-terminal like | Unknown | Florio, Fietz, Pollen | Homo (before Neandertal-Denisovan split) | 6 EGF-like domains [UniProt] |
| SMN2 | Survival of motor neuron 2, centromeric | Loss of SMN1 and SMN2 results in embryonic death; mutations in SMN1 are associated with spinal muscular atrophy, mutations in SMN2 do not lead to disease; forms heteromeric complexes with proteins such as SIP1 and GEMIN4, and also interacts with several proteins known to be involved in the biogenesis of snRNPs, such as hnRNP U protein and the small nucleolar RNA binding protein [RefSeq] | Pollen, Miller | Homo (after Neandertal-Denisovan split) | Evolved after the split from Neanderthal and Denisovan (*Dennis et al., 2017*); telomeric (SMN1) and centromeric (SMN2) copies of this gene are nearly identical and encode the same protein; critical sequence difference between the two genes is a single nucleotide in exon 7, which is thought to be an exon splice enhancer; the full length protein encoded by this gene localizes to both the cytoplasm and the nucleus [RefSeq]; GEMIN2 binding site, tudor domain, RPP20/POP7 interaction site, SNRPB binding site, SYNCRIP interaction site [UniProt] |
| ZNF492 | Zinc finger protein 492 | Unknown | Florio, Fietz, Pollen | Homo (before Neandertal-Denisovan split) | Human ZNF492 is a chimera consisting of the original KRAB repressor domain and the acquired ZNF98 DNA binding domain (this publication); KRAB domain and 13 C2H2 zinc finger motifs [UniProt] |
| ALG1L | ALG1, chitobiosyldiphosphodolichol beta-mannosyltransferase like | Unknown | Pollen, Miller | Hominini | Unknown |
| CBWD2 | COBW domain containing 2 | Unknown | Pollen, Miller | Hominini | CobW domain, ATP binding sites [UniProt] |
| TMEM133 | Transmembrane protein 133 | Unknown | Fietz, Miller, Johnson | Hominini | Intronless gene [RefSeq]; transmembrane protein without signal peptide and two predicted transmembrane domains (Protter) |
| HHLA3 | HERV-H LTR-associating 3 | Unknown | Fietz, Pollen | Homininae | Unknown |
| TMEM99 | Transmembrane protein 99 | Unknown | Fietz, Miller | Hominidae | Transmembrane protein with signal peptide and three transmembrane domains [UniProt, Protter] |
| ZNF90 | Zinc finger protein 90 | Unknown | Florio, Pollen | Hominidae | KRAB domain and 15 C2H2 zinc finger motifs [UniProt] |
| CCDC74B | Coiled-coil domain containing 74B | Unknown | Fietz, Pollen, Miller, Johnson | Hominoidea | Coiled-coil motif [UniProt] |
| C9orf47 | Chromosome nine open reading frame 47 | Unknown | Fietz, Miller, Johnson | Hominoidea | Signal peptide [UniProt, Protter] |

*Table 1 continued*

| Gene symbol | Gene name | Function | cNPC-enriched in | Occurrence | Features |
|---|---|---|---|---|---|
| GLUD2 | Glutamate Dehydrogenase 2 | Localized to the mitochondrion, homohexamer, recycles glutamate during neurotransmission and catalyzes the reversible oxidative deamination of glutamate to alpha-ketoglutarate [RefSeq] | Miller, Johnson | Hominoidae | Arose by retroposition (intronless) (this publication) |
| PTTG2 | Pituitary tumor-transforming 2 | Unknown | Fietz, Miller | Hominoidae | Arose by retroposition; reading frame remained open only in apes (this publication); destruction box, SH3 binding domain [UniProt] |
| APOL2 | Apolipoprotein L2 | Is found in the cytoplasm, where it may affect the movement of lipids or allow the binding of lipids to organelles [RefSeq] | Florio, Fietz, Pollen, Johnson | Catarrhini | Signal peptide [UniProt, Protter] |
| APOL4 | Apolipoprotein L4 | May play a role in lipid exchange and transport throughout the body, as well as in reverse cholesterol transport from peripheral cells to the liver [RefSeq] | Fietz, Miller | Catarrhini | Signal peptide [UniProt, Protter] |
| BTN3A2 | Butyrophilin subfamily three member A2 | Immunoglobulin superfamily, may be involved in the adaptive immune response [RefSeq] | Fietz, Pollen, Miller | Catarrhini | Signal peptide, Ig-like V-type domain, coiled coil motif, one transmembrane domain [UniProt, Protter] |
| BTN3A3 | Butyrophilin Subfamily 3 Member A3 | Major histocompatibility complex (MHC)-associated gene | Fietz, Miller | Catarrhini | Arose by triplication duplication: BTN3A1 is likely the ancestral gene, BTN3A1 duplicated once and this 'copy' duplicated to BTN3A2 and BTN3A3. This triplication happened in the human-rhesus ancestor since marmoset has only a single gene (this publication); type I membrane protein with two extracellular immunoglobulin (Ig) domains and an intracellular B30.2 (PRYSPRY) domain [UniProt] |
| MICA | MHC class I polypeptide-related sequence A | Is a ligand for the NKG2-D type II integral membrane protein receptor; functions as a stress-induced antigen that is broadly recognized by intestinal epithelial gamma delta T cells; variations have been associated with susceptibility to psoriasis one and psoriatic arthritis [RefSeq] | Florio, Fietz, Miller | Catarrhini | Signal peptide, Ig-like C1-type domain, one transmembrane domain [UniProt, Protter] |
| MT1M | Metallothionein 1M | Unknown | Miller, Johnson | Catarrhini | Two metal-binding domains [UniProt] |
| SLFN13 | Schlafen Family Member 13 | Unknown | Florio, Johnson | Catarrhini | Unknown |
| ZNF100 | Zinc finger protein 100 | Unknown | Fietz, Pollen | Catarrhini | KRAB domain and 12 C2H2 zinc finger motifs [UniProt] |
| ZNF222 | Zinc Finger Protein 222 | Unknown | Pollen, Miller | Catarrhini | KRAB domain and 10 C2H2 zinc finger motifs [UniProt] |
| ZNF43 | Zinc finger protein 43 | Unknown | Fietz, Pollen | Catarrhini | KRAB domain and 22 C2H2 zinc finger motifs [UniProt] |
| ZNF695 | Zinc finger protein 695 | Unknown | Florio, Fietz, Miller | Catarrhini | KRAB domain and 13 C2H2 zinc finger motifs [UniProt] |
| ZNF724 | Zinc finger protein 724 | Unknown | Florio, Fietz, Pollen | Catarrhini | KRAB domain and 16 C2H2 zinc finger motifs [UniProt] |
| ZNF726 | Zinc finger protein 726 | Unknown | Florio, Fietz | Catarrhini | KRAB domain and 20 C2H2 zinc finger motifs [UniProt] |

*Table 1 continued*

| Gene symbol | Gene name | Function | cNPC-enriched in | Occurrence | Features |
|---|---|---|---|---|---|
| ZNF730 | Zinc finger protein 730 | Unknown | Fietz, Johnson | Catarrhini | KRAB domain and 12 C2H2 zinc finger motifs [UniProt] |
| ZNF732 | Zinc finger protein 732 | Unknown | Florio, Pollen | Catarrhini | KRAB domain and 16 C2H2 zinc finger motifs [UniProt] |
| ZNF816 | Zinc finger protein 816 | Unknown | Florio, Fietz, Pollen, Miller | Catarrhini | KRAB domain and 15 C2H2 zinc finger motifs [UniProt] |
| ZNF93 | Zinc finger protein 93 | Unknown | Pollen, Miller | Catarrhini | KRAB domain and 17 C2H2 zinc finger motifs [UniProt] |
| HEPN1 | Hepatocellular carcinoma, down-regulated 1 | Transient expression of this gene significantly inhibits cell growth and suggests a role in apoptosis; downregulated or lost in hepatocellular carcinomas [RefSeq] | Florio, Fietz, Miller | Simiiformes | Expressed in the liver; encodes a short peptide, predominantly localized to the cytoplasm [RefSeq] |
| KIF4B | Kinesin family member 4B | A microtubule-based motor protein that plays vital roles in anaphase spindle dynamics and cytokinesis [RefSeq] | Fietz, Pollen | Simiiformes | Intronless retrocopy of kinesin family member 4A [RefSeq]; kinesin motor domain, ATP binding site, coiled coil, nuclear localization signal, PRC1 interaction domain [UniProt] |
| ZNF20 | Zinc finger protein 20 | Unknown | Fietz, Miller | Simiiformes | KRAB domain and 15 C2H2 zinc finger motifs [UniProt] |
| ZNF680 | Zinc finger protein 680 | Unknown | Florio, Pollen | Simiiformes | KRAB domain and 12 C2H2 zinc finger motifs [UniProt] |
| ZNF718 | Zinc finger protein 718 | Unknown | Fietz, Pollen | Simiiformes | KRAB domain and 11 C2H2 zinc finger motifs [UniProt] |
| ZNF788 | Zinc finger family member 788 | Unknown | Fietz, Pollen | Simiiformes | No KRAB domain, 17 C2H2 zinc finger motifs [UniProt] |
| MT1E | Metallothionein 1E | Unknown | Pollen, Miller | Haplorrhini | Two metal-binding domains [UniProt] |
| TNFRSF10D | TNF receptor superfamily member 10d | Does not induce apoptosis and has been shown to play an inhibitory role in TRAIL-induced cell apoptosis. [RefSeq] | Florio, Fietz | Haplorrhini | Signal peptide, TRAIL-binding domain, one transmembrane domain, truncated death domain [UniProt, Protter] |

DOI: https://doi.org/10.7554/eLife.32332.005

we confirmed by genomic PCR (data not shown), resulting in a reading frame shift that predicts a shorter, 52-amino acid-long polypeptide. Taken together, we conclude that the human-specific cNPC-enriched protein-coding genes evolved mainly by entire or partial gene duplications.

## Validation of human-specific gene duplications

We sought to corroborate that the human-specific cNPC-enriched protein-coding genes arising from complete or partial gene duplication indeed constitute additional gene copies (rather than reflecting the inability of distinguishing multiple gene copies in the genomes of the other great apes due to genome assembly issues). To this end, we used a quantitative genomic PCR approach. The rationale was that primers targeting genomic regions within duplicated loci that are identical in human, chimpanzee and bonobo should amplify genomic DNA of the three species proportionally to the copy number of each gene in each species. As a proof of principle, we validated the known human-specific nature of the partially duplicated *ARHGAP11B* by designing primers to the regions that are identical between *ARHGAP11A* and *ARHGAP11B*. Using the bonobo gene as the standard, this resulted in a two-fold increase of the human PCR product compared to the bonobo and chimpanzee, confirming that *ARHGAP11B* is indeed a human-specific gene duplication (*Figure 4E*) and that our approach allows us to estimate copy number variants.

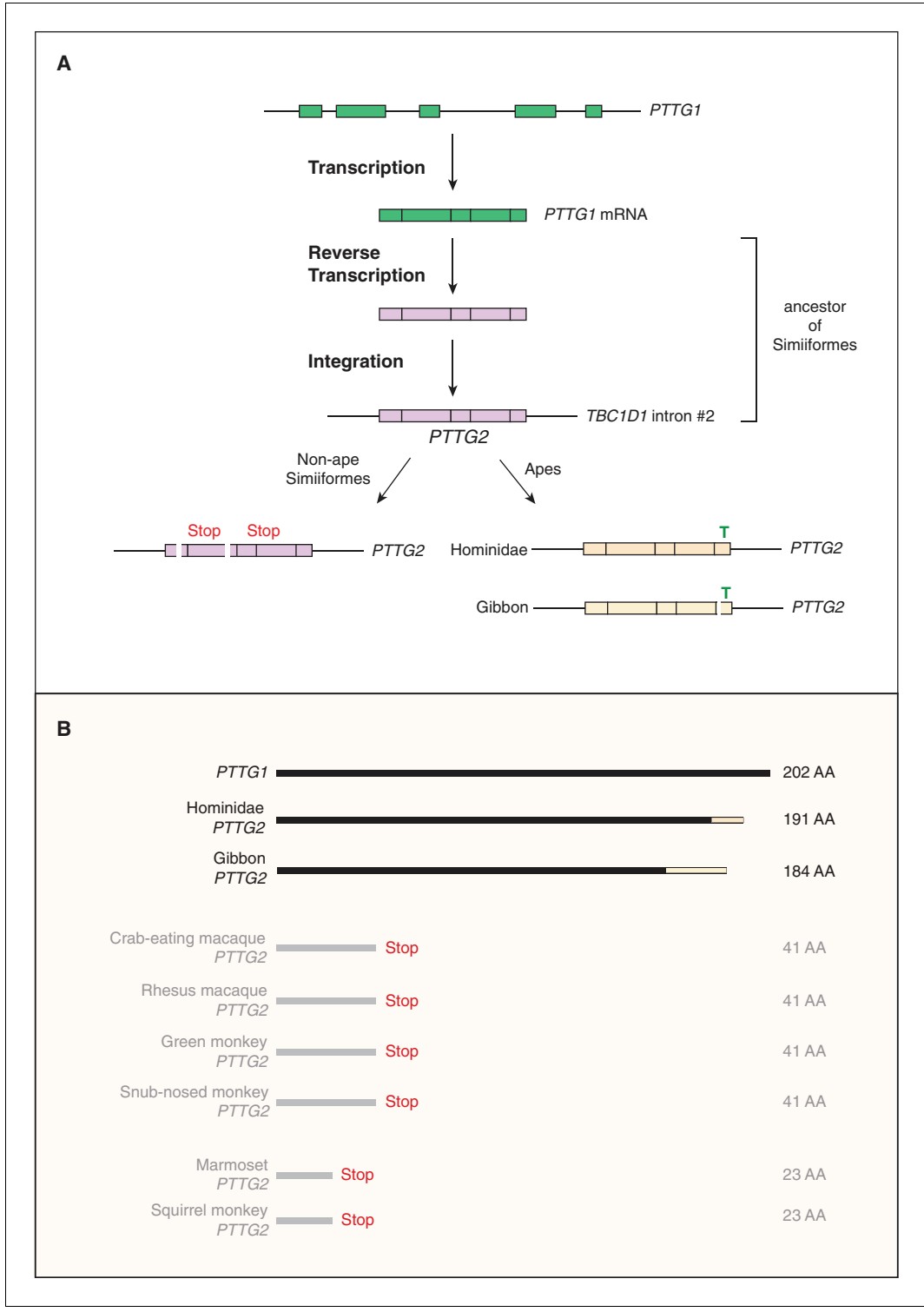

**Figure 3.** Evolutionary origin of the *PTTG2* gene. (**A**) Origin of the *PTTG2* gene by reverse transcription of the *PTTG1* mRNA and insertion as a retroposon into the *TBC1D1* locus in the ancestor to New-World monkeys, Old-World monkeys and apes (Simiiformes). (**B**) Comparison of the PTTG1 and Hominoidea PTTG2 polypeptides, and of the prematurely closed open-reading frames of non-ape Simiiformes *PTTG2*.

DOI: https://doi.org/10.7554/eLife.32332.006

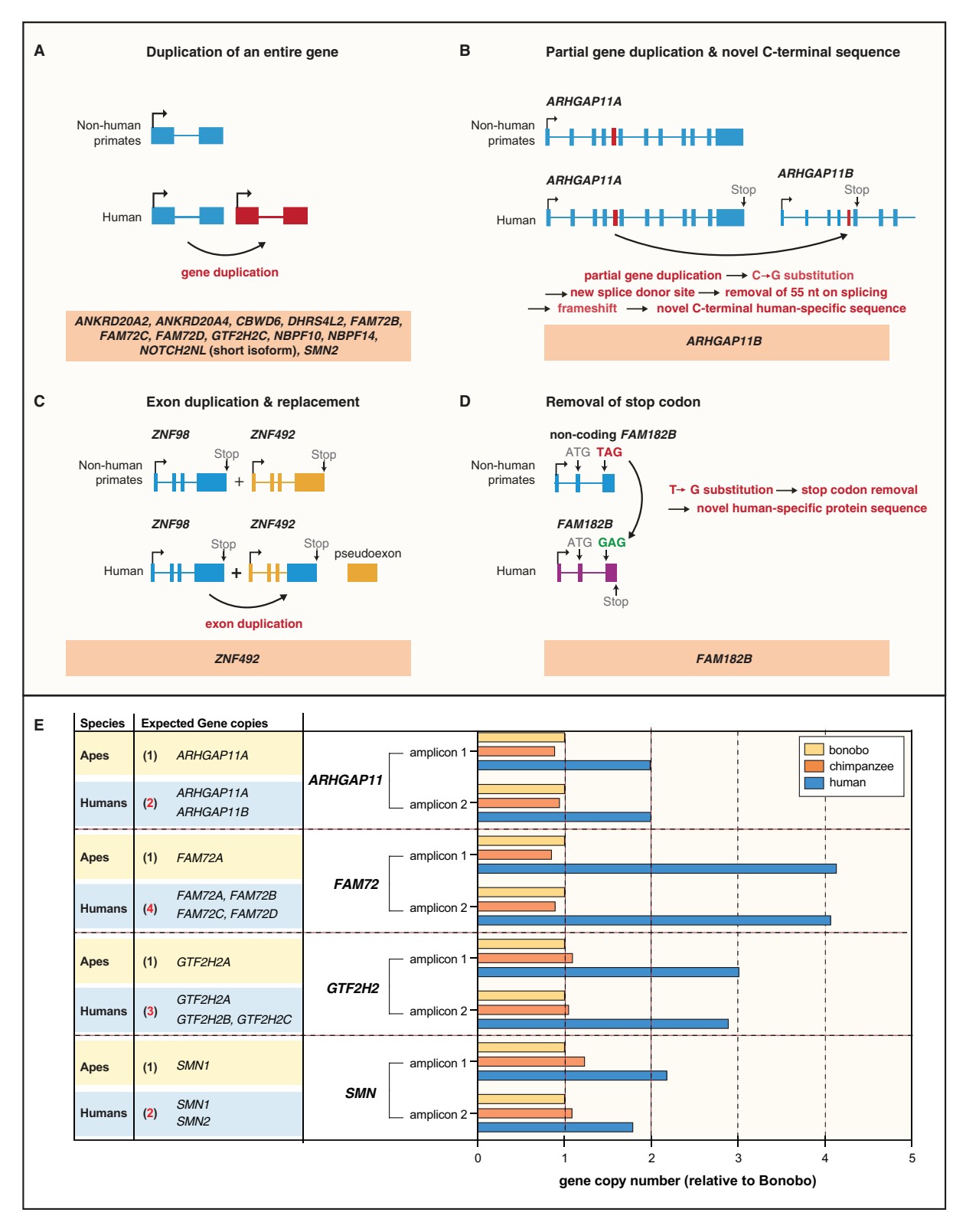

**Figure 4.** Evolution of the human-specific cNPC-enriched protein-coding genes. Diagrams depicting the evolutionary origin of the 15 human-specific genes. (**A**) Duplication of the entire ancestral gene, which applies to 12 of the human-specific genes. *NOTCH2NL* is included in this group because it initially arose by duplication of the entire *NOTCH2* gene. Note that the gene duplication giving rise to *SMN2* occurred after the Neandertal – modern human lineage split, whereas the other 11 gene duplications occurred before that split (**Dennis et al., 2017**). (**B**) Partial gene duplication giving rise to

*Figure 4 continued*

*ARHGAP11B* ~ 5 Mya (**Riley et al., 2002**; **Antonacci et al., 2014**; **Dennis et al., 2017**). Note that a single C–>G substitution in exon 5 (red box), which likely occurred after the gene duplication event but before the Neandertal – modern human lineage split, created a new splice donor site, causing a reading frame shift that resulted in a novel, human-specific 47 amino acid C-terminal sequence (**Florio et al., 2015**; **Florio et al., 2016**). (C) Exon duplication and replacement giving rise to human *ZNF492*. Exon 4 of *ZNF98* (blue) is duplicated and inserted into intron 3 of *ZNF492* (orange), rendering the original *ZNF492* exon 4 a pseudoexon. (D) Removal of a stop codon converting the non-coding *FAM182B* of non-human primates into the protein-coding human *FAM182B*. A single T–>G substitution removes the stop codon at the 5' end of exon 3, thereby creating an open reading frame (purple). (E) Validation of the human-specific nature of selected human genes by determination of their copy numbers. Human (blue), chimpanzee (orange) and bonobo (yellow) genomic DNA was used as template to perform a qPCR that would generate two distinct amplicons of both, the gene common to all three species (black regular letters) and the human-specific gene(s) under study (red bold letters), as indicated. The relative amounts of amplicons obtained for each of the four gene groups are depicted with the amounts of amplicons obtained with the bonobo genomic DNA as template being set to 1.0. Note that compared to chimpanzee and bonobo genomic DNA, the copy number in human genomic DNA is (i) two-fold higher for *ARHGAP11*, consistent with the presence of the human-specific gene *ARHGAP11B* in addition to the common gene *ARHGAP11A*; (ii) four-fold higher for *FAM72*, consistent with the presence of the human-specific genes *FAM72B*, *FAM72C* and *FAM72D* in addition to the common gene *FAM72A*; (iii) three-fold higher for *GTF2H2*, consistent with the presence of the human-specific genes *GTF2H2B* (black bold letters, not among the cNPC-enriched genes identified in this study) and *GTF2H2C* in addition to the common gene *GTF2H2A*; and (iv) two-fold higher for *SMN*, consistent with the presence of the human-specific gene *SMN2* in addition to the common gene *SMN1*.

DOI: https://doi.org/10.7554/eLife.32332.007

The following source data and figure supplements are available for figure 4:

**Source data 1.** Human raw data.
DOI: https://doi.org/10.7554/eLife.32332.010
**Source data 2.** Bonobo raw data.
DOI: https://doi.org/10.7554/eLife.32332.011
**Source data 3.** Chimpanzee raw data.
DOI: https://doi.org/10.7554/eLife.32332.012
**Figure supplement 1.** Evolution of *NOTCH2NL*.
DOI: https://doi.org/10.7554/eLife.32332.008
**Figure supplement 2.** Validation of the genomic qPCR specificity.
DOI: https://doi.org/10.7554/eLife.32332.009

We therefore used the same approach to validate other human-specific genes in our list arising from complete gene duplication. For 7 of these 12 genes (*ANKRD20A2*, *ANKRD20A4*, *CBWD6*, *DHRS4L2*, *NBPF10*, *NBPF14*, and *NOTCH2NL*), we could not design primers that uniquely target these genes as the respective genomic loci are not well resolved in the non-human great ape genomes. Thus, the final validation of these putative human-specific genes awaits improved genome assemblies. For the other five human-specific cNPC-enriched genes (*FAM72B/C/D*, *GTF2H2C*, *SMN2*) and for the human-specific gene *GTF2H2B* for which primers could be designed, genomic qPCR resulted in an estimated four human copies of *FAM72* (i.e. the ancestral *FAM72A* plus its three human-specific paralogs *FAM72B/C/D*), three human copies of *GTF2H2* (i.e. the ancestral *GTF2H2* and the two human-specific paralogs *GTF2H2B/C*) and two human copies of *SMN* (i.e. ancestral *SMN1* and its human-specific paralog *SMN2*) (**Figure 4E**), compared to only one copy in both chimpanzee and bonobo. This validated the human-specific nature of these genes.

To validate the specificity of the genomic qPCR reactions, we sequenced the amplicons of the above-mentioned seven human-specific genes and their ancestral paralogs from human, bonobo, and chimpanzee genomic DNA. The percentage of the DNA sequence reads of the PCR amplicons that aligned to the targeted genomic sequences ranged from 97.2 to 99.8, indicating that the PCR reactions were highly specific (**Figure 4—figure supplement 2A**). Moreover, the absolute number of DNA sequence reads that aligned to the respective genomic sequence of either one of the seven human-specific genes or its ancestral paralog (**Figure 4—figure supplement 2B**) corresponded to the gene copy numbers as determined by the genomic qPCR (**Figure 4E**). These two sets of data therefore validated the genomic qPCR data.

## Spatial mRNA expression analysis in fetal human neocortex of selected primate- and human-specific cNPC-enriched genes

Given the 15 human-specific genes that had emerged from our screen for cNPC-enriched genes, it was of interest to examine their spatial expression pattern in the various zones of the fetal human

cortical wall. To this end, we performed in-situ hybridization (ISH) on 13 wpc human neocortex for 13 of the 15 human-specific cNPC-enriched genes, and for the three above-described primate- but not human-specific genes, to determine the localization of their mRNAs.

We were able to design specific ISH probes for six human-specific genes – *ARHGAP11B*, *NOTCH2NL*, *DHRS4L2*, *FAM182B*, *GTF2H2C* and *ZNF492*. In the case of *ARHGAP11B*, we used a specific Locked Nucleic Acid (LNA) probe, which enabled us to distinguish the mRNA of *ARHGAP11B* from that of *ARHGAP11A* (*Figure 5—figure supplement 1*). mRNA expression was detected in all three germinal zones (VZ, iSVZ and oSVZ) but not in the CP (*Figure 5B*). Expression of *NOTCH2NL* was essentially restricted to the VZ (*Figure 5D*). *DHRS4L2* and *ZNF492* were found to be expressed in all three germinal zones and the CP, with a stronger signal in the VZ and iSVZ than in the oSVZ and CP (*Figure 5F,I*). *GTF2H2C* mRNA expression was also detected in the three germinal zones and the CP, but with stronger staining in the VZ, iSVZ and CP than in the oSVZ (*Figure 5H*). Finally, in the case of *FAM182B*, mRNA expression was stronger in the VZ and CP than in the iSVZ and oSVZ (*Figure 5J*).

We then sought to compare expression of these human-specific genes with that of their ancestral paralog. We were able to design probes specific to *ARHGAP11A* (ancestral to *ARHGAP11B*) and *NOTCH2* (ancestral to *NOTCH2NL*). The mRNA expression pattern of the ancestral paralog *ARHGAP11A* (*Figure 5A*) showed a striking difference to that of the human-specific gene *ARHGAP11B* (*Figure 5A,B*). Whereas *ARHGAP11B* expression was largely restricted to the three germinal zones (*Figure 5B*), *ARHGAP11A* expression was also detected in the IZ and to some extent the CP (*Figure 5A*). Of note, *ARHGAP11A*, in contrast to *ARHGAP11B*, showed a specific ISH signal at the basal surface of the CP, which likely reflects concentration of *ARHGAP11A* mRNA in the basal end-feet of radial glial cells (*Figure 5A,B*), a subcellular site at which certain mRNAs can be concentrated (*Tsunekawa et al., 2012*; *Pilaz et al., 2016*). In contrast to *ARHGAP11A/B*, mRNA localization of ancestral *NOTCH2* (*Figure 5C*) was virtually identical to the expression pattern of human-specific *NOTCH2NL* described above (*Figure 5C,D*).

We could not design ISH probes specific to the ancestral copies of *DHRS4L2* (i.e. *DHRS4*) and *GTF2H2C* (i.e. *GTF2H2*). We therefore designed probes that recognize all paralogs within a given gene family and detect their combined mRNA expression. Specifically, for *DHRS4*, the ancestral paralog, the human-specific *DHRS4L2*, and the other paralog, *DHRS4L1*; for *GTF2H2*, the ancestral paralog, the cNPC-enriched human-specific *GTF2H2C*, and the human-specific, but not cNPC-enriched, *GTF2H2B*. The combined mRNA expression patterns of *DHRS4/L1/L2* were similar to that of *DHRS4L2*, and the combined mRNA expression patterns of *GTF2H2/B/C* were similar to that of *GTF2H2C*, with stronger signal in the VZ and iSVZ than in the oSVZ and CP in both cases (*Figure 5E–H*).

In the case of *ANKRD20A2*, *ANKRD20A4*, *CBWD6*, *FAM72B*, *FAM72C*, *FAM72D*, and *SMN2*, for which we could not design specific probes, we also could not design probes specific to their ancestral paralogs. We therefore used probes recognizing all paralogs in each family and analyzed their combined expression patterns. In the case of *ANKRD20A1-4* (which included the cNPC-enriched human-specific genes *ANKRD20A2* and *ANKRD20A4*, ancestral *ANKRD20A1*, and yet another paralog, *ANKRD20A3*), expression was essentially confined to the VZ (*Figure 5K*). In the case of *CBWD1-6* (which included the cNPC-enriched human-specific gene *CBWD6*, ancestral *CBWD1*, and *CBWD2-5*), mRNA expression was stronger in the VZ and CP than iSVZ and oSVZ (*Figure 5L*). In the case of *FAM72A-D* (which included the cNPC-enriched human-specific genes *FAM72B*, *FAM72C*, and *FAM72D* and ancestral *FAM72A*), mRNA expression was stronger in the VZ and iSVZ than in the oSVZ and CP (*Figure 5M*). A similar expression pattern was found for *SMN1-2* (which included the cNPC-enriched human-specific gene *SMN2* and ancestral *SMN1*) (*Figure 5N*).

Finally, we also used ISH to examine the spatial expression pattern in the fetal human cortical wall of the three primate-specific genes *PTTG2*, *MICA* and *KIF4B*. Due to the high degree of similarity in nucleotide sequence this analysis also included the mRNA of the respective ancestral paralog. mRNA expression for *PTTG1/2* (*Figure 6A*), *MICA/B* (*Figure 6B*) and *KIF4A/B* (*Figure 6C*) was robust in the human VZ and iSVZ, relatively low in the oSVZ, and moderate in the CP.

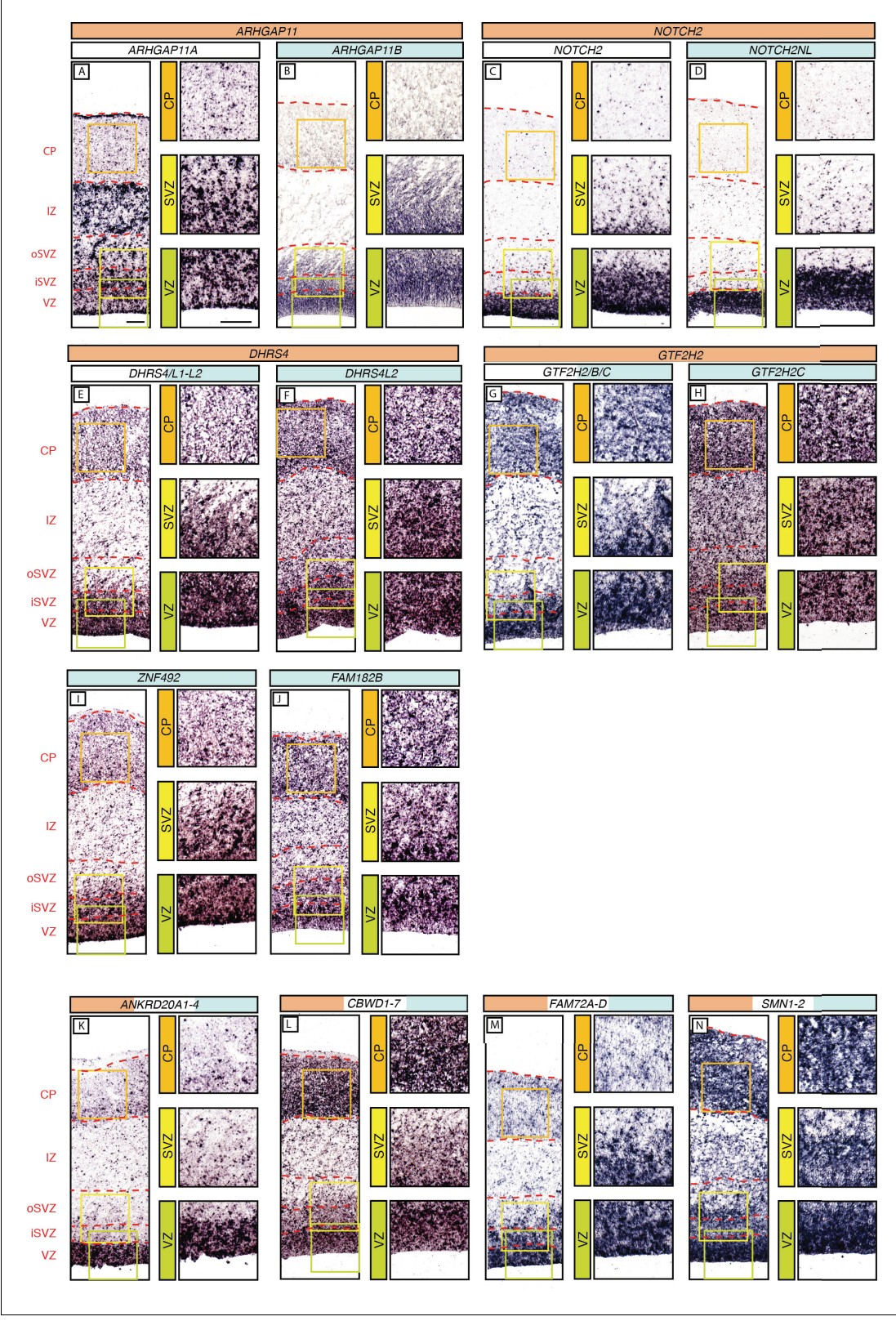

**Figure 5.** In-situ hybridization analysis of the mRNA levels of the human-specific cNPC-enriched protein-coding genes in the various zones of the fetal neocortical wall. Coronal sections of human fetal neocortex (13 wpc) were subjected to ISH using probes that (i) are specific for the mRNA of the human-specific gene under study (B, D, F, H, I, J), indicated by the gene name with blue background; (ii) recognize the mRNAs of both the human-specific gene(s) and the paralog gene(s) common to other primates as well (E, G, K, L, M, N), indicated by gene names with white/blue background; or
*Figure 5 continued on next page*

*Figure 5 continued*

(iii) are specific to the ancestral paralog (**A**, **C**), indicated by the gene name with white background. The various zones of the fetal neocortical wall are indicated on the left and by red dashed lines. Green, yellow, and orange boxes indicate areas of the VZ, SVZ and CP, respectively, that are shown at higher magnification in the respective images on the right. Scale bars in A apply to all panels and are 100 µm. Note that an ISH probe yielding a reliable signal for *ZNF98* could not be designed.

DOI: https://doi.org/10.7554/eLife.32332.013

The following figure supplement is available for figure 5:

**Figure supplement 1.** *ARHGAP11B*-specific ISH probe.

DOI: https://doi.org/10.7554/eLife.32332.014

## Cell type-specific expression patterns of the human-specific cNPC-enriched protein-coding genes compared to the corresponding ancestral paralogs

Complete or partial gene duplications often encompass the regulatory elements that control gene expression (*Bailey et al., 2002*; *Eichler et al., 2004*; *Fortna et al., 2004*; *Hurles, 2004*). This raises the question whether the human-specific cNPC-enriched protein-coding genes identified here exhibit similar cell-type expression patterns as their respective ancestral paralogs, or whether expression differences have evolved during human evolution.

Given that by ISH we could not distinguish the majority of the human-specific genes from their respective ancestral paralog, we sought an additional approach to gain insight into potential differences in expression between ancestral and human-specific paralogs. Specifically, we used our previously reported cell-type-specific RNA-Seq data from the human aRG population (aRG), the bRG population (bRG) and the neuron fraction (N) (*Florio et al., 2015*) and re-analyzed these data using Kallisto. Kallisto is a probabilistic algorithm to estimate absolute transcript abundance, which has been proven to be accurate in assigning RNA-Seq reads to specific transcripts, including those originating from highly similar paralog genes (*Bray et al., 2016*). We could confidently ascertain cell-type-specific mRNA expression profiles for 12 of the 15 human-specific genes and their corresponding ancestral paralog (Figure 7).

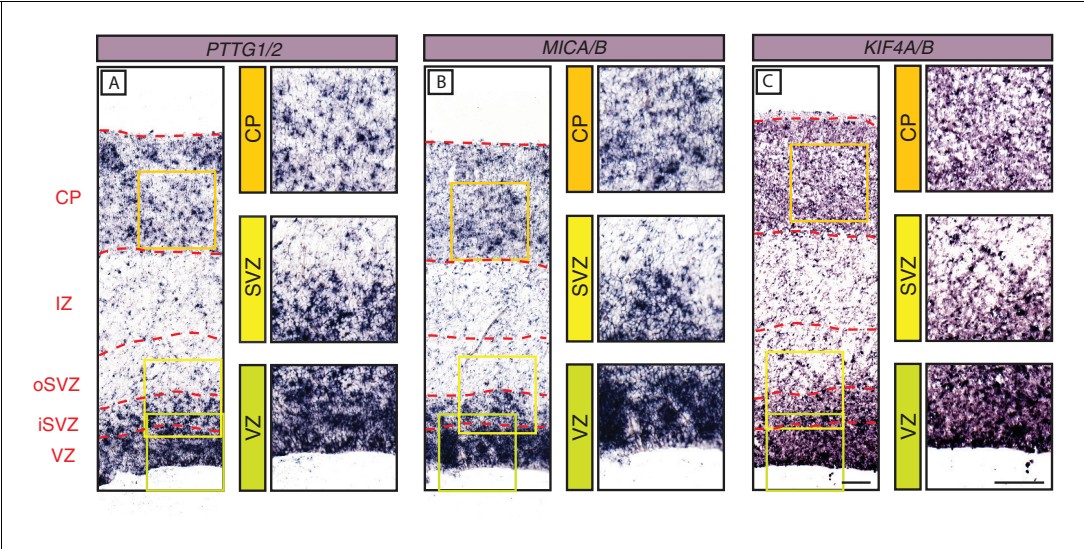

**Figure 6.** In-situ hybridization analysis of the mRNA levels of three selected primate-specific genes in the various zones of the fetal human neocortical wall. Coronal sections of human fetal neocortex (13 wpc) were subjected to ISH using probes recognizing the mRNAs of the primate-specific genes *PTTG2* (**A**), *MICA* (**B**) and *KIF4B* (**C**) and their ancestral paralogs *PTTG1* (**A**), *MICB* (**B**), and *KIF4A* (**C**). The various zones of the fetal neocortical wall are indicated on the left and by red dashed lines. Green, yellow, and orange boxes indicate areas of the VZ, SVZ, and CP, respectively, that are shown at higher magnification in the respective images on the right. Scale bars in C apply to all panels and are 100 µm.

DOI: https://doi.org/10.7554/eLife.32332.015

We first focused on changes in total mRNA levels between the human-specific genes and their ancestral paralogs. With the exception of *NOTCH2NL,* for which the total mRNA levels in aRG, bRG, and N were in the same range as the corresponding ancestral paralog, we found that the majority of the human-specific genes showed markedly different total mRNA expression levels compared to their ancestral paralog, which were either reduced (*ARHGAP11B*, *CBWD6*, *FAM72B/C/D*, *GTF2H2C*, *SMN2*) or increased (*ANKRD20A2*, *ANKRD20A4*, *DHRS4L2*, *ZNF492*) (*Figure 7B*). This reflects either changes in mRNA expression levels per cell, changes in the proportions of mRNA-expressing cells, or both. Irrespective of which is the case, this finding indicates that certain features of expression of these human-specific genes in the cNPC-to-neuron lineage might have changed compared to their ancestral paralogs during human evolution. This in turn raises the possibility that with the appearance of these human-specific genes their roles in the cell types concerned may have undergone some modification.

Next, we asked whether the human-specific genes diverged in their pattern of expression in aRG vs. bRG vs. N from that of their ancestral paralogs. For five of the human-specific genes (*CBWD6*, *FAM72B/C/D*, and *NOTCH2NL*), the pattern of mRNA levels in these three cell populations was similar to that of the respective ancestral paralog (*Figure 7A*). In the case of the other seven human-specific genes, we observed differences in the expression in aRG vs. bRG. For instance, *ZNF492* expression is lower in bRG than aRG, in contrast to what is observed for ancestral *ZNF98* expression. Vice versa, *ARHGAP11B*, *DHRS4L2*, *GTF2H2C*, and *SMN2* expression is higher in bRG than aRG, in contrast to what is observed for the respective ancestral paralog. Of note, the increase in the *ARHGAP11B* mRNA level in bRG as compared to aRG is consistent with the previously reported function of this gene in basal progenitor amplification (*Florio et al., 2015*; *Florio et al., 2016*). Moreover, we observed decreases (*ANKRD20A2*, *ANKRD20A4*) or increases (*DHRS4L2*, *GTF2H2C*) in the N fraction mRNA level, in relation to the aRG and bRG mRNA levels, compared to their respective ancestral paralog (*Figure 7A*). These findings suggest that these six human-specific genes underwent changes in regulatory elements at the transcriptional and/or post-transcriptional level.

We sought to corroborate these data by subjecting the previously prepared cDNAs of aRG, bRG, and N (*Florio et al., 2015*) to qPCR analysis to quantify the transcripts of selected human-specific genes and their respective ancestral paralogs. We could design appropriate qPCR primers for four such pairs, *ARHGAP11B* vs. *ARHGAP11A*, *GTF2H2C* vs. *GTF2H2*, *NOTCH2NL* vs. *NOTCH2*, and *ZNF492* vs. *ZNF98*. As shown in *Figure 7—figure supplement 1*, this analysis largely confirmed the results of the Kallisto analysis (*Figure 7A*).

To complement these data, we performed a second type of analysis. We identified paralog-specific sequencing reads (*Figure 7—source data 1*; see *Figure 7—figure supplement 2A* for illustration of a hypothetical example) using our previously reported RNA-Seq dataset (*Florio et al., 2015*), and then determined the number of paralog-specific sequencing reads for the 11 human-specific genes and their corresponding ancestral paralog in aRG, bRG, and N (*Figure 7—figure supplement 2B*). This analysis largely corroborated the results shown in *Figure 7A*, further pointing to expression changes in aRG vs. bRG vs. N for the human-specific genes in comparison to their corresponding ancestral paralogs.

We next ascertained expression of all human-specific genes identified in this study across all cell fractions (Florio, Johnson), single cells (Pollen) and cortical layers (Fietz, Miller) in the five gene sets analyzed (*Figure 7—figure supplements 3,4*). Moreover, when this information was available, we determined expression of each gene in each individual fetal sample used to build these datasets (*Figure 7—figure supplements 3,4C*), thus providing information about inter-individual variation. This analysis revealed good congruence between all gene sets (*Figure 7—figure supplements 3,4*), and showed that virtually all genes analyzed were expressed in all individual specimens studied (with the exception of *NBPF10/14* in the Florio dataset, where these genes were not detected altogether; *Figure 7—figure supplement 3*).

In addition, since the Fietz dataset sampled six different fetal samples from four distinct gestational ages (*Figure 7—figure supplement 3*), we could investigate the temporal progression of expression of these genes during corticogenesis. While the majority of these genes show relatively constant or fluctuating gene expression levels across stages, some (*NOTCH2NL*, *FAM182B*) are enriched in the germinal zones at 13 wpc (early neurogenesis) compared to 14–16 wpc (mid-neurogenesis), while others (e.g. *FAM72C*, *FAM72D*) show the opposite expression pattern, suggesting that these genes may have differential roles during corticogenesis.

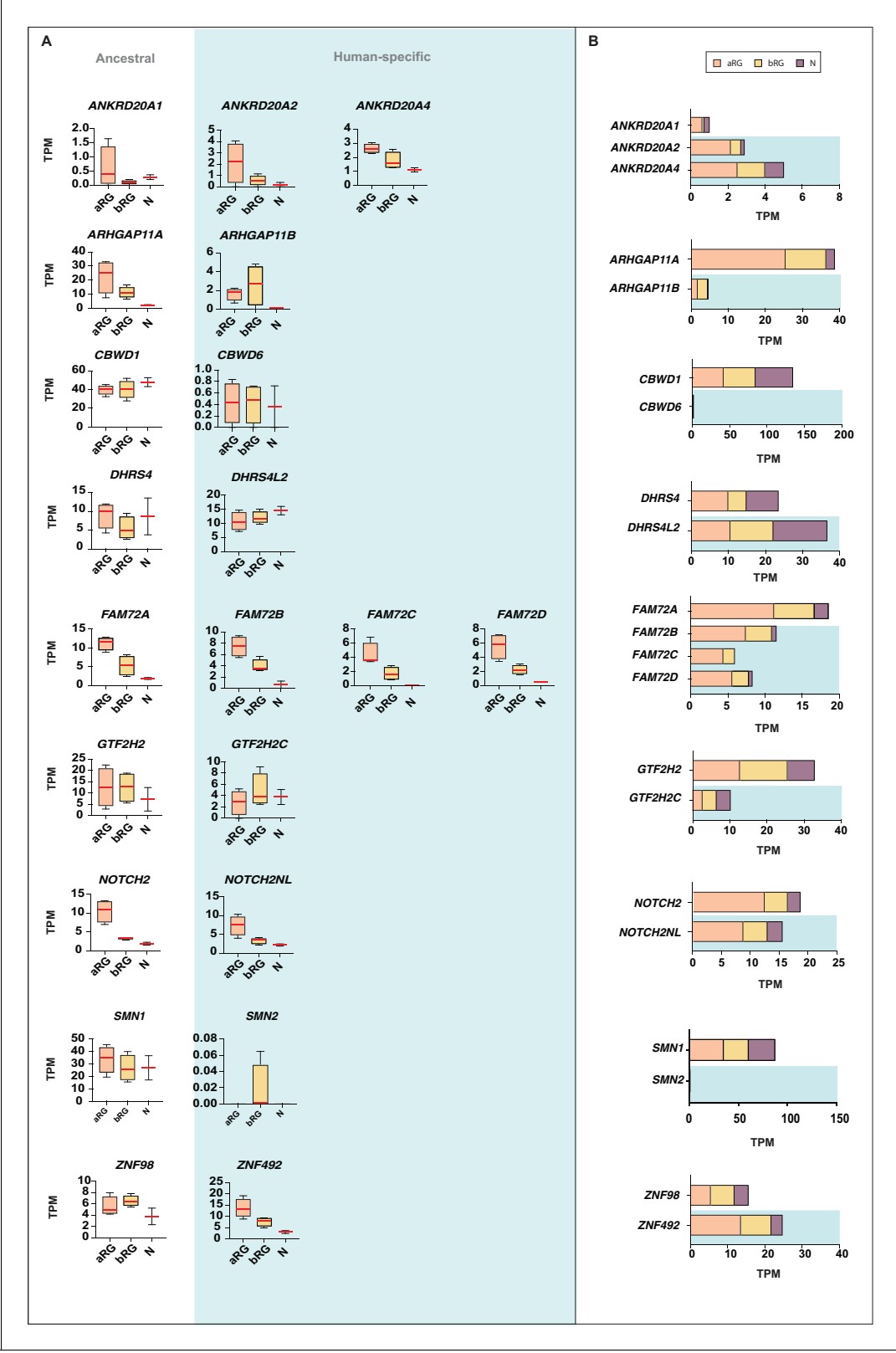

**Figure 7.** Comparison of the mRNA expression of 12 human-specific cNPC-enriched protein-coding genes with their ancestral paralogs in isolated cell populations enriched in aRG, bRG and neurons from fetal human neocortex. A previously published genome-wide transcriptome dataset obtained by RNA-Seq of cell populations isolated from fetal human neocortex, that is, aRG (orange) and bRG (yellow) in S-G2-M and a fraction enriched in neurons but also containing bRG in G1 (N, purple) (***Florio et al., 2015***), was analyzed for the abundance of mRNA-Seq reads assigned to either the indicated

*Figure 7 continued on next page*

*Figure 7 continued*

human-specific gene(s) under study (blue background) or the corresponding ancestral paralog (white background), using the Kallisto algorithm. (**A**) Min-max box-and-whiskers plots showing mRNA levels (expressed in Transcripts Per Million, TPM); red lines indicate the median. (**B**) Stacked bar plots showing the cumulative mRNA expression levels in the indicated cell types (sum of the median TPM values shown in (**A**)).

DOI: https://doi.org/10.7554/eLife.32332.016

The following source data and figure supplements are available for figure 7:

**Source data 1.** Alignments of the mRNA sequences of ancestral and human-specific paralogs of the orthology groups ANKRD20A, ARHGAP11, CBWD, DHRS4, FAM72, GTF2H2, NOTCH2 and ZNF98.

DOI: https://doi.org/10.7554/eLife.32332.021

**Figure supplement 1.** qPCR validation of the Kallisto analysis.

DOI: https://doi.org/10.7554/eLife.32332.017

**Figure supplement 2.** Comparison of the paralog-specific mRNA expression between 11 human-specific cNPC-enriched genes and their respective ancestral paralog in aRG, bRG and neuron-enriched cell populations from fetal human neocortex.

DOI: https://doi.org/10.7554/eLife.32332.018

**Figure supplement 3.** mRNA expression levels of the 15 human-specific, cNPC-enriched, protein-coding genes in the human individuals analyzed in the Fietz et al., Florio et al. and Johnson et al. transcriptome datasets.

DOI: https://doi.org/10.7554/eLife.32332.019

**Figure supplement 4.** Analysis of the expression of the 15 human-specific, cNPC-enriched, protein-coding genes in the cell types of the Pollen et al. transcriptome dataset and in the cortical zones of the Miller et al. transcriptome dataset.

DOI: https://doi.org/10.7554/eLife.32332.020

We finally explored the complexity in cell-type-specific expression patterns by examining the differential mRNA expression of protein-coding splice variants of the human-specific genes. Specifically, we analyzed our aRG vs. bRG vs. N RNA-Seq data (*Florio et al., 2015*) for cell-type-specific gene expression and relative abundance of sequencing reads diagnostic of specific protein-coding splice variants of 14 of the 15 human-specific cNPC-enriched genes (*Figure 8*, *Supplementary file 4*). This showed, for most of these human-specific genes (*ANKRD20A2*, *ANKRD20A4*, *CBWD6*, *DHRS4L2*, *FAM72B/C*, *FAM182B*, *GTF2H2C*, *NBPF14*, *NOTCH2NL*, *SMN2*), the preferential expression of certain splice variants. Moreover, this analysis revealed splice variants with preferential expression in either aRG or bRG for some of these human-specific genes (e.g. *ARHGAP11B*, *CBWD6*, *GTF2H2C*, *SMN2*). A notable case was *ARHGAP11B*, of which one splice variant (Ensembl transcript ENST00000428041), endowed with a shorter 3'-UTR, was exclusively expressed in bRG whereas the other splice variant (Ensembl transcript ENST00000622744) was enriched in aRG (*Figure 8*).

In summary, these analyses show that after duplication, the expression pattern of most of the resulting new, human-specific cNPC-enriched protein-coding genes evolved differences in both the levels and cell-type specificity of their mRNAs compared to their respective ancestral paralog.

## Human-specific *NOTCH2NL* promotes basal progenitor proliferation

To illustrate the value of our resource for studying potential roles of the human-specific cNPC-enriched genes in neocortical expansion, we focused on *NOTCH2NL*, as Notch signaling is known to be important for cNPC behavior (*Kawaguchi et al., 2008*; *Pierfelice et al., 2008*; *Lui et al., 2011*; *Imayoshi et al., 2013*; *Wilkinson et al., 2013*). The NOTCH2NL protein is equivalent to the NOTCH2 protein encoded by the shortest *NOTCH2* splice variant (*Figure 4—figure supplement 1*). To investigate a potential role of *NOTCH2NL* in cNPCs, we used in utero electroporation to express *NOTCH2NL* under the control of a constitutive promoter in neocortical aRG of embryonic day (E) 13.5 mouse embryos. Analysis of the progeny of the targeted aRG by immunofluorescence for PCNA and Ki67, two markers of cycling cells, 48 hr after *NOTCH2NL* electroporation revealed an increase in cycling basal progenitors in the SVZ and IZ, but not in apical progenitors in the VZ (*Figure 9A–D*). This increase involved bIPs expressing Tbr2 (*Figure 9F,G*), rather than bRG expressing Sox2 (*Figure 9H,I*). Concomitant with this increase, we observed a decrease in cell cycle exit in the SVZ, as indicated by the decrease in Ki67 expression in the progeny of BrdU-labeled cells derived from the targeted aRG (*Figure 9C,E*). These data were further corroborated by analysis of mitotic cNPCs using phosphohistone H3 immunofluorescence, which showed an increase in

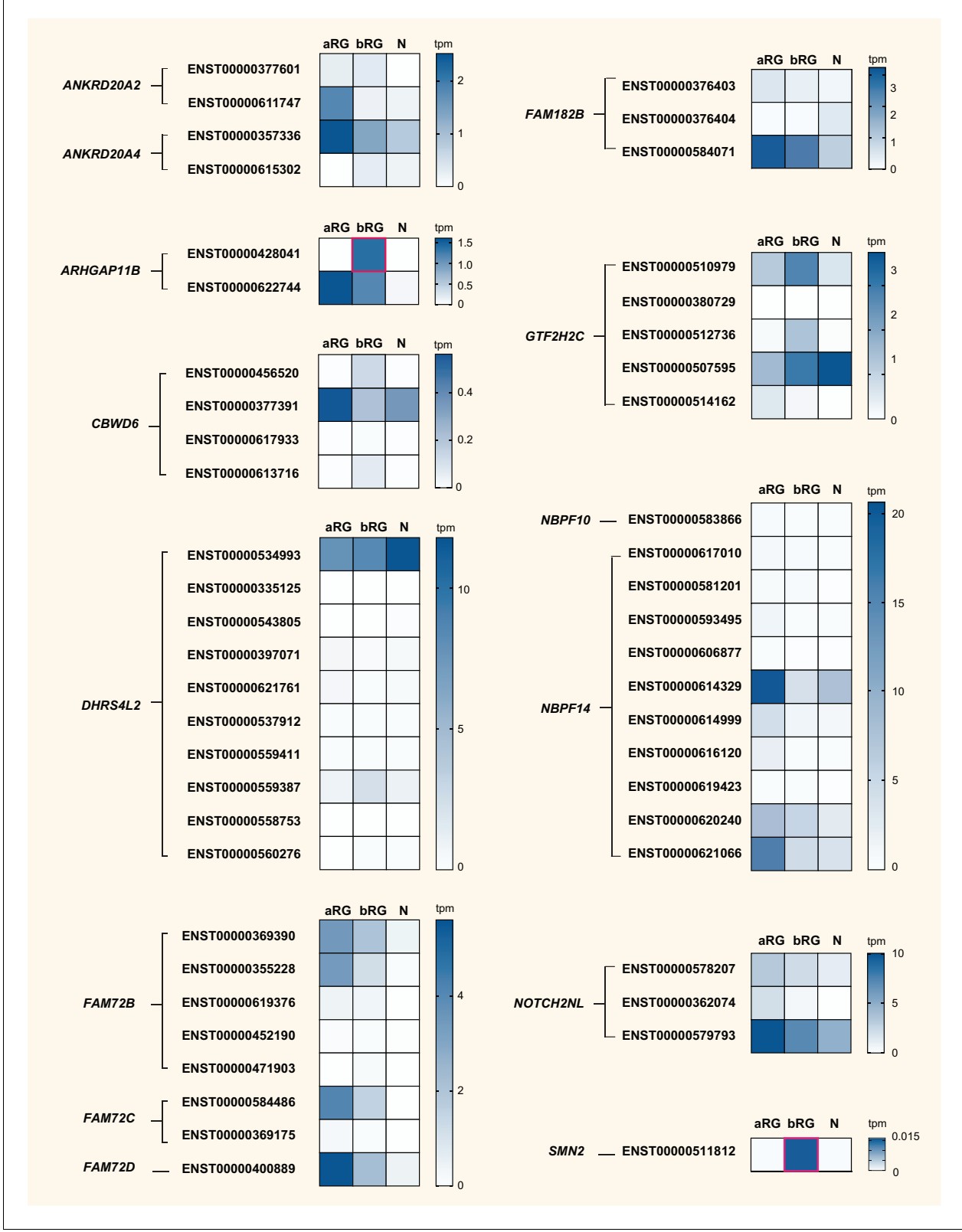

**Figure 8.** Cell-type specificity of mRNA expression of splice variants encoded by 14 human-specific cNPC-enriched genes. Heatmaps showing TPM expression levels (see color keys on right) of all protein-coding splice variants encoded by the indicated 14 human-specific cNPC-enriched genes in aRG, bRG and neuron-enriched (N) cell populations from fetal human neocortex (*Florio et al., 2015*). Only splice variants with detectable expression, albeit very low in some cases, are shown. *ZNF492* is not shown as only one splice variant exists. See *Supplementary file 4* for mRNA expression data

*Figure 8 continued on next page*

*Figure 8 continued*

for each cell type and splice variant, including non-coding transcripts. Human-specific genes are grouped based on orthology, and splice variants (indicated by Ensembl transcript IDs) encoded by the respective cNPC-enriched human-specific gene(s) are grouped together. Note that ENST00000428041, a splice variant of *ARHGAP11B* and ENST00000511812, a splice variant of *SMN2*, are uniquely expressed in bRG (red boxes). Splice variant-specific mRNA expression was assessed using the Kallisto algorithm.

DOI: https://doi.org/10.7554/eLife.32332.022

abventricular, but not ventricular, mitoses (*Figure 9J,K*). Thus, forced expression of the human-specific *NOTCH2NL* gene in mouse embryonic neocortex promotes basal progenitor proliferation.

## Discussion

Our study not only provides a resource of genes that are promising candidates to exert specific roles in the development and evolution of the primate, and notably human, neocortex, but also has implications regarding (i) the emergence of these genes during primate evolution and (ii) the maintenance vs. modification of the cell-type specificity of their expression.

As to the emergence of these genes during primate evolution, two aspects of our findings deserve comment. First, different mechanisms contributed to the origin of human-specific cNPC-enriched protein-coding genes. While entire or partial gene duplications gave rise to the vast majority of these genes, consistent with previous results (*Bailey et al., 2002*; *Eichler et al., 2004*; *Fortna et al., 2004*; *Hurles, 2004*), we found that exon duplications can give rise to chimeric genes, as observed for *ZNF492*, and that the removal of a translational stop codon can create a new open reading frame, as observed for *FAM182B*. Among the primate- but not human-specific genes, two genes with functions that are likely relevant for cell proliferation (*KIF4B*, *PTTG2*; (*Brosius, 1991*; *Long et al., 2003*; *Marques et al., 2005*)) arose by retroposition of a reverse transcribed spliced mRNA, highlighting another mechanism for the emergence of new genes. Here, *PTTG2* is a particularly interesting case, since it was inactivated during the evolution of non-Hominoidea Simiiformes but remained intact during the evolution of Hominoidea. This raises the possibility that *PTTG2* may exert a role in the development of the neocortex of apes and human but not of New-World and Old-World monkeys. Given the expression of *PTTG2* in the germinal zones of fetal human neocortex and the fact that this gene is derived from *PTTG1*, which encodes a protein exhibiting tumorigenic activity (*Vlotides et al., 2007*), it will be interesting to explore whether *PTTG2* may amplify cNPCs.

Second, of the 50 primate-specific human genes, 15 (30%) are human-specific and 17 (34%) arose in the Catarrhini ancestor. These percentages are higher than expected from a constant rate of gene emergence on the phylogenetic branches leading from the primate ancestor to human. Indeed, relating the number of new cNPC-enriched genes to the rate of neutral mutations revealed the two branches leading to Catarrhini (branch 6, *Figure 2A*) and to human (branch 1, *Figure 2A*), respectively, as outliers (*Figure 2B*). With regard to the branch leading to Catarrhini, one may speculate whether the increased appearance of new cNPC-enriched genes was related to the concomitant increase in gyrencephaly. With regard to the branch leading to human, one may speculate whether the increased appearance of new cNPC-enriched genes was related to the concomitant increase in brain size.

As to the issue of maintenance vs. modification of the cell-type specificity of expression of the human-specific genes, it is striking to observe that the majority of these genes, despite arising by entire or partial gene duplications, show marked differences not only in the level but also in the cNPC-type specificity of their mRNA expression compared to their ancestral paralog. For several of the human-specific genes, the corresponding spatial characteristics of their mRNA expression in the neocortical germinal zones and cNPC types could be corroborated by specific ISH and cell-type-specific RNA-Seq data, respectively. These data suggest that during human evolution, after gene duplication, these genes underwent specific changes in regulatory elements at the transcriptional and/or post-transcriptional level.

Our resource of primate-specific genes provides promising candidates that could have contributed to the evolution of primate-specific, including human-specific, features of neocortical development. Indeed, we found that expressing the human-specific cNPC-enriched gene *NOTCH2NL* in mouse embryonic neocortex increased the abundance of cycling basal progenitors, a hallmark of the

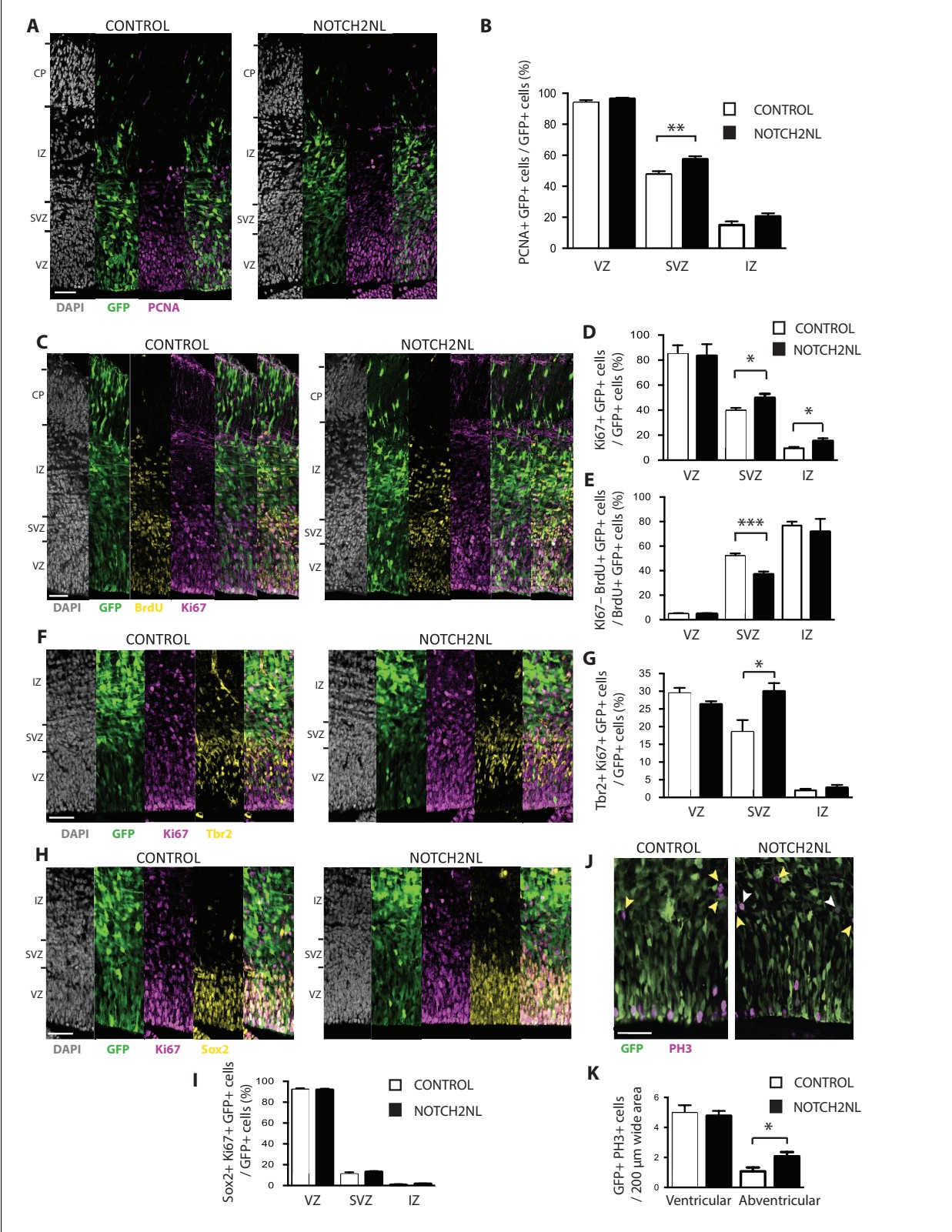

**Figure 9.** Forced expression of *NOTCH2NL* in mouse embryonic neocortex increases cycling basal progenitors. The neocortex of E13.5 mouse embryos was in utero co-electroporated with a plasmid encoding GFP together with either an empty vector (Control) or a *NOTCH2NL* expression plasmid (NOTCH2NL), all under constitutive promoters, followed by analysis 48 hr later. Bromodeoxyuridine (BrdU) was administered by intraperitoneal injection (10 mg/kg) into pregnant mice at E14.5 (**C, E**). (**A**) GFP (green) and PCNA (magenta) double immunofluorescence combined with DAPI staining

*Figure 9 continued on next page*

*Figure 9 continued*

(white) of control (left) and *NOTCH2NL*-electroporated (right) neocortex. (B) Quantification of the percentage of the progeny of the targeted cells, that is, the GFP+ cells, that are PCNA+ in the VZ, SVZ and IZ upon control (white columns) and *NOTCH2NL* (black columns) electroporation. (C) GFP (green), BrdU (yellow), and Ki67 (magenta) triple immunofluorescence combined with DAPI staining (white) of control (left) and *NOTCH2NL*-electroporated (right) neocortex. (D) Quantification of the percentage of the progeny of the targeted cells, that is, the GFP+ cells, that are Ki67+ in the VZ, SVZ, and IZ upon control (white columns) and *NOTCH2NL* (black columns) electroporation. (E) Quantification of the percentage of the BrdU-labeled progeny of the targeted cells, that is, the GFP+ cells, that are Ki67–, that is, that did not re-enter the cell cycle, in the VZ, SVZ, and IZ upon control (white columns) and *NOTCH2NL* (black columns) electroporation. (F, H) GFP (green), Ki67 (magenta), and either Tbr2 (F) or Sox2 (H) (yellow) triple immunofluorescence combined with DAPI staining (white) of control (left) and *NOTCH2NL*-electroporated (right) neocortex. (G, I) Quantification of the percentage of the progeny of the targeted cells, that is, the GFP+ cells, that are Ki67+ and Tbr2+ (G) or Ki67+ and Sox2+ (I) in the VZ, SVZ and IZ upon control (white columns) and *NOTCH2NL* (black columns) electroporation. (J) GFP (green) and phosphohistone H3 (PH3, magenta) double immunofluorescence of control (left) and *NOTCH2NL*-electroporated (right) neocortex. Yellow arrowheads, GFP– and PH3+ abventricular cells. White arrowheads, GFP+ and PH3+ abventricular cells. (K) Quantification of the number of ventricular and abventricular progeny of the targeted cells, that is, the GFP+ cells, that are in mitosis (PH3+) in a 200 µm-wide microscopic field upon control (white columns) and *NOTCH2NL* (black columns) electroporation. (A, C, F, H, J) Images are single 2 µm optical sections. Scale bars, 50 µm. (B, D, E, G, I, K) Data are mean of 6–11 embryos each, averaging the numbers obtained from 1 to 4 cryosections per embryo (one 100 µm-wide (B, D, E, G, I) or 200 µm-wide (K) microscopic field per cryosection). Error bars indicate SEM; *p<0.05; **p<0.01;***p<0.001; Student's t-test.
DOI: https://doi.org/10.7554/eLife.32332.023

developing human neocortex. The NOTCH2NL protein studied here is predicted to lack a signal peptide, which raises the issue of whether the NOTCH2NL protein is secreted, and if so, via which pathway, or whether the effect of *NOTCH2NL* in basal progenitors is due to the *NOTCH2NL* mRNA only or to an action of the NOTCH2NL protein in the cytoplasm.

Moreover, we previously showed that the human-specific function of *ARHGAP11B* in cNPCs arose by a single-nucleotide substitution that generated a new splice donor site, the use of which generates a novel human-specific C-terminal protein sequence that we implicate in basal progenitor amplification (*Florio et al., 2015*; *Florio et al., 2016*). Importantly, this single-nucleotide substitution presumably occurred relatively recently during human evolution (*Florio et al., 2016*), that is, after the partial gene duplication event ~5 million years ago (*Riley et al., 2002*; *Antonacci et al., 2014*; *Dennis et al., 2017*). Furthermore, we have identified here an *ARHGAP11B* splice variant that is specifically expressed in human bRG (*Figure 8*), the basal progenitor type thought to have a key role in neocortex expansion (*Lui et al., 2011*; *Borrell and Reillo, 2012*; *Betizeau et al., 2013*; *Borrell and Götz, 2014*; *Florio and Huttner, 2014*). Interestingly, in contrast to the other protein-coding *ARHGAP11B* splice variant detected, which contains a long 3'-UTR with predicted microRNA binding sites and which is predominantly expressed in aRG, the bRG-specific *ARHGAP11B* splice variant contains only a short 3'-UTR lacking predictable microRNA binding sites. This suggests that *ARHGAP11B* mRNAs may be subject to differential, microRNA-mediated, regulation depending on whether ARHGAP11B functions in the lineage progression from aRG to bRG or in bRG amplification. Taken together, our findings reveal genomic changes at a variety of levels that gave rise to novel functions and patterns of expression in cNPCs and that are likely relevant for the development and evolution of the human neocortex.

# Materials and methods

## Key resources table

| Reagent type (species) or resource | Designation | Source or reference | Identifiers | Additional information |
|---|---|---|---|---|
| Strain, strain background (*Mus musculus*) | C57BL/6J | MPI-CBG Animal Facility | | |
| Biological sample (*Homo sapiens*) | fetal neocortex tissue (13 wpc) | Universitätsklinikum Carl Gustav Carus Dresden | | |
| Antibody | anti-BrdU (mouse) | MPI-CBG Antibody Facility | | (1:1000) |
| Antibody | anti-GFP (chicken polyclonal) | Abcam | Abcam Cat# ab13970, RRID:AB_300798 | (1:1000) |

*Continued on next page*

*Continued*

| Reagent type (species) or resource | Designation | Source or reference | Identifiers | Additional information |
|---|---|---|---|---|
| Antibody | anti-PH3 (rat monoclonal) | Abcam | Abcam Cat# ab10543, RRID:AB_2295065 | (1:1000) |
| Antibody | anti-Tbr2 (mouse) | MPI-CBG Antibody Facility | | (1:500) |
| Antibody | anti-Sox2 (goat polyclonal) | R + D Systems | R and D Systems Cat# AF2018, RRID:AB_355110 | (1:500) |
| Antibody | anti-Ki67 (rabbit polyclonal) | Abcam | Abcam Cat# ab15580, RRID:AB_443209 | (1:500) |
| Antibody | anti-PCNA (mouse monoclonal) | Millipore | Millipore Cat# CBL407, RRID:AB_93501 | (1:500) |
| Antibody | Alexa Fluor 488-, 555- and 594-secondaries | Molecular Probes | | (1:500) |
| Recombinant DNA reagent | pCAGGS | doi: 10.1126/science.aaa1975 | | |
| Recombinant DNA reagent | pCAGGS-GFP | doi: 10.1126/science.aaa1975 | | |
| Recombinant DNA reagent | pCAGGS-NOTCH2NL | this paper | | *NOTCH2NL* was PCR amplified from cDNA and cloned into pCAGGS |
| Sequence-based reagent | *ARHGAP11B* LNA probe | this paper | | AGTCTGGTACACGCCCTTCTTTTCT |
| Sequence-based reagent | *DHRS4L2* LNA probe | this paper | | AGACAGTGGCGGTTGCGTGA |
| Sequence-based reagent | *FAM182B* LNA probe | this paper | | GCAGGGATACACGGCTAT |
| Sequence-based reagent | *GTF2H2C* LNA probe | this paper | | TCAGACGGCCTGCC |
| Software, algorithm | cutadapt (v1.15) | https://cutadapt.readthedocs.io/en/stable/ | RRID:SCR_011841 | |
| Software, algorithm | STAR (v2.5.2b) | https://github.com/alexdobin/STAR | RRID:SCR_015899 | |
| Software, algorithm | Bedtools | http://bedtools.readthedocs.io/en/stable/# | RRID:SCR_006646 | |
| Software, algorithm | R | The R Foundation | | |
| Software, algorithm | samtools | Genome Research Limited | RRID:SCR_002105 | |
| Software, algorithm | bowtie1 | http://bowtie-bio.sourceforge.net/index.shtml | RRID:SCR_005476 | |
| Software, algorithm | BioMart | Bioconductor | | |
| Software, algorithm | BLAT | http://genome.ucsc.edu/cgi-bin/hgBlat?command=start | RRID:SCR_011919 | |
| Software, algorithm | Kallisto | doi:10.1038/nbt.3519 | | |
| Software, algorithm | FastQC | Babraham Bioinformatics | RRID:SCR_014583 | |
| Software, algorithm | dupRadar | Bioconductor | | |
| Software, algorithm | DESeq2 | Bioconductor | RRID:SCR_015687 | |
| Software, algorithm | GeneTrail2 | https://genetrail2.bioinf.uni-sb.de | | |
| Other | CESAR | doi: 10.1093/nar/gkw210 | | |

## Human fetal brain tissue

Human fetal brain tissue was obtained from the Klinik und Poliklinik für Frauenheilkunde und Geburtshilfe, Universitätsklinikum Carl Gustav Carus of the Technische Universität Dresden, following elective termination of pregnancy and informed written maternal consent, and with approval of the local University Hospital Ethical Review Committees. The gestational age of the specimen used for ISH (13 wpc) was assessed by ultrasound measurements of crown-rump length, as described previously (*Florio et al., 2015*). Immediately after termination of pregnancy, the tissue was placed on ice and transported to the lab. The sample was then transferred to ice-cold Tyrode's solution, and tissue

fragments of cerebral cortex were identified and dissected. Tissue was fixed in 4% paraformalde-hyde in 120 mM phosphate buffer (pH 7.4) for 3 hr at room temperature followed by 24 hr at 4°C. Fixed tissue was then incubated in 30% sucrose overnight, embedded in Tissue-Tek OCT (Sakura, Netherlands), and frozen on dry ice. Cryosections of 12 µm were produced using a cryostat (Microm HM 560, Thermo Fisher Scientific) and stored at –20°C until processed for ISH.

## Mice

All animal experiments were performed in accordance with German animal welfare laws and over-seen by the institutional review board. C57BL/6J mice were maintained in specific pathogen-free conditions in the MPI-CBG animal facility.

## Identification of human cNPC-enriched protein-coding genes

To identify genes that are preferentially expressed in human cNPCs, five published transcriptome datasets (*Fietz et al., 2012*; *Florio et al., 2015*; *Johnson et al., 2015*; *Miller et al., 2014*; *Pollen et al., 2015*) were screened as described below; these transcriptome data had been gener-ated from 13 to 21 wpc human fetal neocortex, using diverse cortical zone or cell-type-enrichment strategies and modes of determination of RNA levels (summarized in *Supplementary file 1*).

*Fietz et al. (2012)* – This transcriptome dataset (*Fietz et al., 2012*) was generated by RNA-Seq of the neocortical germinal zones (VZ, iSVZ, oSVZ) and CP isolated by LCM from the neocortex of 6 human fetuses ranging in gestational age from 13 to 16 wpc. The data were screened for protein-coding genes more highly expressed, across all stages, in either VZ, iSVZ, or oSVZ than CP (as deter-mined by DGE analysis, $p<0.01$ and FPKM$\geq$1.5). The resulting gene set contained 2780 genes (*Supplementary file 1*, *Figure 1*).

*Miller et al. (2014)* (BrainSpan Atlas of the Allen Brain Institute, Prenatal LMD Microarray, http://www.brainspan.org/lcm/search/index.html) – This transcriptome dataset (*Miller et al., 2014*) was generated by microarray RNA expression profiling of germinal zones (VZ, iSVZ, oSVZ) and neuron-enriched layers (IZ, subplate, CP, marginal zone, subpial granular zone) isolated by LCM from fetal human neocortex. The data were screened for protein-coding genes with highest correlation with either VZ, iSVZ, or oSVZ (correlation coefficient >0.25) compared to all neocortical regions analyzed. Correlation scores were taken from the original publication (see *Supplementary file 1*). For the pur-pose of this analysis, we paired together correlation scores from the two 15–16 wpc samples and the two 21 wpc samples originally included in the study. A gene was considered to be cNPC-enriched if it showed laminar correlation with either of the three germinal zones in both 15–16 wpc samples or in both 21 wpc samples. The resulting gene set contained 3802 genes (*Supplementary file 1*, *Figure 1*).

*Florio et al. (2015)* – This transcriptome dataset (*Florio et al., 2015*) was generated by RNA-Seq of human radial glia subtypes (aRG and bRG) and CP neurons (N) isolated from the neocortex of two 13 wpc human fetuses. These cell types were differentially labeled using a combination of fluores-cent molecular markers, and isolated by FACS. By experimental design, only cells that exhibited api-cal plasma membrane and/or contacted the basal lamina were isolated. Moreover, the isolation of aRG and bRG was confined to cells that had duplicated their DNA, and the neuron fraction con-tained a minority of bRG in G1 (*Florio et al., 2015*). The data were screened for protein-coding genes with higher expression in either aRG or bRG than N (as determined by DGE analysis, $p<0.01$ and FPKM$\geq$0.5). For this analysis, we used differential analysis data from the original publication (see *Supplementary file 1*). The resulting gene set contained 2030 genes (*Supplementary file 1*, *Figure 1*).

*Pollen et al. (2015)* – This transcriptome dataset (*Pollen et al., 2015*) was generated by RNA-Seq of single cells captured from the VZ and SVZ microdissected from the neocortex of three 16–18 wpc human fetuses. Cells were post-hoc attributed – based on gene expression profiling – to either radial glia (aRG and bRG), intermediate progenitors (i.e. bIPs), or neurons (N). The data were screened for genes positively correlated with either radial glia or bIPs (correlation coefficient >0.1) and negatively correlated with N (correlation coefficient <0.03). An expression cutoff was set to a minimum of 9 cells with detectable expression for a given gene (i.e. 2% of all sampled cells in the original study). Correlation scores were taken from the original publication (see *Supplementary file 1*). The resulting gene set contained 4391 genes (*Supplementary file 1*, *Figure 1*).

*Johnson et al. (2015)* – This transcriptome dataset (*Johnson et al., 2015*) was generated by RNA-Seq of human radial glia subtypes (aRG and bRG) and a population of intermediate progenitors and neurons isolated from the neocortex of one 18 wpc and two 19 wpc human fetuses. These cell types were differentially labeled using a combination of fluorescent molecular markers, and isolated by FACS. For the purpose of the present analysis, the original RNA-Seq data were re-processed as follows: sequencing reads were checked for overall quality using FastQC (v0.11.2). Read alignments were performed using human genome reference assembly GRCh38 and quantification of genes of Ensembl release v88 was done using STAR (v2.5.2b). Duplicated reads were identified using Picard MarkDuplicates (v2.10.2) and were analyzed with dupRadar (v1.6.0). Differential gene expression analysis on raw counts was performed with DESeq2 (v1.16.1). Data were screened for protein-coding genes with higher expression in aRG and/or bRG as compared to the cell population enriched in intermediate progenitors and neurons (as determined by DGE analysis, p<0.01 and FPKM$\geq$0.1). The resulting gene set contained 1617 genes (*Supplementary file 1*, *Figure 1*).

The gene sets resulting from these analyses contain only protein-coding genes, which were identified and selected using the Ensembl data-mining tool BioMart (http://www.ensembl.org/biomart/martview/), implementing the Genome Reference Consortium Human Build 38 (GRCh38.p10) dataset.

Next, the five gene sets obtained were intersected. To do this, all gene IDs contained in the five original transcriptome datasets were converted to match the latest Ensembl gene annotation (Ensembl v89) of the GRCh38.p10 genome assembly. The five gene sets obtained were then searched for the co-occurrence of genes (or lack thereof). This resulted in 3458 human cNPC-enriched protein-coding genes present in at least two of the five gene sets (listed in *Supplementary file 1*, see also *Figure 1*).

## Gene ontology (GO) term enrichment analysis

GO term enrichment analysis was performed using GeneTrail2 (https://genetrail2.bioinf.uni-sb.de/) using the 3458 human cNPC-enriched protein-coding genes as input. We performed over-representation analysis as set-level statistic, using the Benjamini-Yekutieli false discovery method to adjust p-values, a significance threshold of 0.05. Raw output of this analysis is shown in *Supplementary file 2*.

## Screening of human cNPC-enriched protein-coding genes for primate-specific orthologs

The 3458 human cNPC-enriched protein-coding genes were screened for the occurrence of one-to-one orthologs in non-primate species, using BioMart and implementing v89 Ensembl annotation of '1-to-1 orthologs'. All genes that had an annotated one-to-one ortholog in non-primate species were excluded from the list of the 3458 human genes. This yielded 77 genes that were candidates to be primate-specific.

Whole genome alignments were visualized in the UCSC genome browser (*Tyner et al., 2017*) to manually analyze each of the 77 candidate primate-specific genes. To this end, co-linear chains of local alignments (*Kent et al., 2003*) between the human hg38 genome assembly and the assemblies of non-primate mammals were inspected to check if the human gene locus aligned to non-primate mammals. For the genes that aligned to non-primate mammals, regardless of whether they aligned in a conserved or in a different context, gene annotations of the aligning species were used to assess which gene is annotated in the respective locus. For this purpose, gene annotations from Refseq, Ensembl (*Aken et al., 2017*) and CESAR (*Sharma et al., 2016*) (a method that transfers human gene annotations to other aligned genomes if the gene has an intact reading frame) were used, and those candidate genes that likely have an aligning gene in non-primate mammals were removed. This reduced the list of the 77 candidates to 50 genes that were considered as primate-specific.

## Tracing the evolution of the primate-specific genes in the primate lineage

The evolution of these 50 primate-specific genes was traced in the primate lineage to determine which of these have orthologs, in non-human primates, to the corresponding 50 human cNPC-enriched protein-coding genes, and which do not, and therefore are human-specific. To this end, co-

linear alignment chains and a multiple genome alignment that includes 17 non-human primate genomes (*Sharma and Hiller, 2017*) were inspected. For the genes that aligned to other primates, the CESAR annotations were used to check if a gene of interest has an intact reading frame in other species. A gene was considered to be conserved only if an intact reading frame is present in the respective species. For example, while *FAM182B* aligns in a conserved context to chimpanzee and gorilla, CESAR did not find an intact reading frame and did not annotate the gene; indeed, inspecting the multiple genome alignment revealed a frameshift in chimpanzee and a stop codon mutation in gorilla, showing that *FAM182B* is likely a non-coding gene in non-human primates. Then, each gene was assigned to a node in the primate phylogeny (clade), based on the descending species that likely have an intact coding gene. Note that this inferred ancestry does not imply that all descending species have an intact gene. This is exemplified by *TMEM99*, which aligns to all great apes and has an intact reading frame in human and orangutan, but encodes no or a truncated protein in chimpanzee/bonobo (due to a frameshift mutation) and gorilla (due to a stop codon mutation).

This analysis was combined with BLAT searches using the human protein or human mRNA sequence to assess the number of aligning loci in other primates; however, this was not conclusive for highly complex loci such as the duplications involving *ANKRD20A* and *CBWD* genes, where numerous similar genes and pseudogenes are present and the completeness of non-human primate genome assemblies is not certain due to the presence of assembly gaps. In addition, for human-specific candidates that arose by duplication, inspecting the respective genomic locus in the chimpanzee genome browser was useful, since human duplications are visible as additional, overlapping alignment chains.

## Paralog-specific and isoform-specific gene expression

To estimate expression differences among cNPC types between (a) given human-specific gene(s) and its/their highly similar ancestral paralog(s) in the human genome, the Kallisto probabilistic algorithm was used, which has been proven to be accurate in assigning reads to specific transcripts, including those originating from highly similar paralog genes in the human genome (*Bray et al., 2016*).

For this analysis, reads generated previously by RNA-Seq of human aRG, bRG and N (SRA Access, SRP052294, (*Florio et al., 2015*)) were used as input, GRCh38 as genome reference, and Ensembl v89 as genome annotation reference. Transcript abundances were output in Transcripts per Million (TPM) units. To compare expression between human-specific and ancestral paralog genes (*Figure 7*), TPM values were extracted for all paralogs in each orthologous group, and the TPM values were summed for all protein-coding transcripts (as per Ensembl annotation) for each gene. To compare expression between different splice variants produced by each human-specific gene (*Figure 8*), the TPM values specific for each individual splice variant were extracted and the data were expressed relative to each other.

Kallisto's transcript abundance measurements represent a probabilistic approximation of actual transcript levels, and thus are an estimate. In order to compare actual paralog gene expression in distinct cNPC types and neurons, a second type of analysis was performed, which did not aim at providing an estimate of absolute transcript abundances, but rather at providing a precise determination of the relative gene expression differences between paralogs. To this end, mRNA sequences of ancestral and human-specific paralogs in each orthology group were aligned, using Clustalw2 (http://www.ebi.ac.uk/Tools/msa/clustalw2/), and the homologous (but not identical) core sequence of each alignment was identified manually (*Figure 7—source data 1*; see *Figure 7—figure supplement 2A* for illustration of a hypothetical example). The corresponding sequences of each paralog – of same length by design – were used as reference for previously generated RNA-Seq reads from aRG, bRG, and N (SRA Access, SRP052294, (*Florio et al., 2015*)) in order to search for paralog-specific mRNA reads. Reads aligning to both, ancestral and human-specific paralogs, were discarded as ambiguous, and only those reads aligning to paralog-specific sites (SNPs or indels), referred to as paralog-specific reads, were used for quantification (*Figure 7—figure supplement 2B*). This stringent alignment was carried out using bowtie1 (bowtie -Sp 5 -m 1 v0).

It should be noted that, in contrast to the Kallisto-based analysis, the latter type of analysis does not distinguish between reads that originate from protein-coding and non-protein-coding transcripts of a given gene. Therefore, the quantifications shown in *Figure 7—figure supplement 2B* reflect

counts of all reads mapping to a given gene, whereas the quantifications shown in *Figure 7* reflect summed counts of protein-coding gene transcripts only.

## qPCR validation of the paralog-specific gene expression analysis

Neocortex of three 12–13 wpc human fetuses, obtained as described above, was used. The isolation of aRG and bRG in S-G2-M and of a fraction enriched in neurons but also containing bRG in G1 from fetal human neocortex, and the preparation of cDNA from these cell populations has already been described (*Florio et al., 2015*). The cDNA libraries obtained from these FACS-isolated fractions were re-analyzed by qPCR, performed as previously described (*Florio et al., 2015*; *Albert et al., 2017*). Primer sequences are provided in *Supplementary file 5*. The qPCR data obtained were normalized to expression of *GAPDH*, as described previously (*Florio et al., 2015*; *Albert et al., 2017*).

## Genomic qPCR

Genomic DNA was obtained from EBV-transformed B cells of human, bonobo and chimpanzee, as described previously (*Prüfer et al., 2012*). Primers (*Supplementary file 6*) were designed for two different amplicons per orthologous gene group to bind to the same region of the human-specific gene(s) under study, its human paralog(s), and the chimpanzee and bonobo orthologs. Only one mismatch in the primer binding sequence between the reference genomes of the three species was allowed.

qPCR was performed on human, chimpanzee and bonobo genomic DNA, using either the ABsolute qPCR SYBR Greenmix (Thermo Fisher Scientific) on a Mx3000P qPCR System (Stratagene) or the Fast Start Essential DNA Green Master (Roche) on a Lightcycler 96 (Roche). The relative copy number between the three species was determined by the comparative cycle threshold (Ct) approach (*Livak and Schmittgen, 2001*) as follows. The Ct values for the human, chimpanzee and bonobo genes under study were normalized to the Ct value of the highly conserved single-copy gene *STX12*. The normalized values were then compared between the three species, using bonobo as reference, to determine the relative copy number.

## Sequencing of genomic PCR products

PCR was performed on human, chimpanzee and bonobo genomic DNA using the REDTaq DNA polymerase (Sigma). Identical cycling and temperature conditions as used for the genomic qPCR described above (annealing at 60°C, 30 cycles) were applied. Input was 15 ng gDNA per 50 μl PCR reaction. Amplicons were checked on 3% agarose gels for the specificity of the PCR reaction; all PCR reactions yielded only one specific band of the correct size. For deep sequencing, amplicons were quantified and pooled at equal molarity such that (i) each pool was specific for one of the three species, and (ii) amplicons targeting an identical locus were distributed into two different pools (see *Figure 4—figure supplement 2A*).

Six barcoded Illumina libraries were generated by ligation of the Illumina-specific sequencing adaptors. Illumina sequencing of these libraries was performed on the MiSEQ device, sequencing regime was $2 \times 150$ bp.

Paired-end data (for raw data, please see *Figure 4—source datas 1–3*) were trimmed using cutadapt (v1.15; -m 20 -q 25 -a file:${Ill_ADAPTERS} -A file:${Ill_ADAPTERS}) and mapped with STAR (v2.5.2b; —alignSJoverhangMin 100 —outFilterType BySJout —sjdbGTFfile ${gtfFile}). Bedtools intersect (v2.25.0) was used to determine the number of overlapping alignments at each locus of interest, and samtools flagstat was used to determine the library size. Final data integration and visualization was implemented using R. The analysis scripts can be found in the repository https://git.mpi-cbg.de/scicomp/Florio_et_al_2018_Validation_of_genomic_qPCR_data.

## *NOTCH2NL* expression in mouse embryonic neocortex

In utero electroporation (30 V, six 50-msec pulses with 1 s intervals) of E13.5 mouse embryos was performed on C57BL/6J mice as described previously (*Florio et al., 2015*), using either 1 μg/μl of pCAGGS-NOTCH2NL and 0.5 μg/μl of pCAGGS-GFP or 1 μg/μl of empty pCAGGS and 0.5 μg/μl of pCAGGS-GFP in PBS containing 0.1% Fast Green. Electroporated neocortex was analyzed at E15.5 following 4% paraformaldehyde fixation (*Florio et al., 2015*). Bromodeoxyuridine (BrdU) was administered by intraperitoneal injection (10 mg/kg) into pregnant mice at E14.5.

## Immunofluorescence

Immunofluorescence was performed on 20 µm cryosections as described previously (*Florio et al., 2015*). The following primary antibodies were used: PCNA, mouse, Millipore, CBL407, 1/500; Ki67, rabbit, Abcam, Ab15580, 1/500; Sox2, goat, R + D Systems, AF2018, 1/500; Tbr2, mouse, MPI-CBG Antibody Facility, 1/500; PH3, rat, Abcam, Ab10543, 1/1000; GFP, chicken, Abcam, Ab13970, 1/1000. For BrdU immunofluorescence, cryosections were quenched for 15 min in 0.1 M glycine in PBS, incubated for 45 min in 2N HCl at 37°C, blocked in 10% horse serum in PBS, and then incubated with a conjugated fluorescent BrdU antibody (mouse, MPI-CBG Antibody Facility, 1/500) for 2 hr at room temperature. The following secondary antibodies were used: Alexa Fluor 488, 555, and 594, Molecular Probes, 1/500.

## In-situ hybridization

Templates were amplified by PCR (see *Supplementary file 7* for primer sequences) from oligo-dT-primed cDNA prepared from fetal human neocortex total RNA, and RNA probes directed against the mRNA(s) of a given human-specific gene and (if applicable) its paralog(s) in the human genome were synthesized using the DIG RNA labeling Mix (Roche). The *ARHGAP11B* LNA probe was designed with the Custom LNA mRNA Detection Probe design tool (QIAGEN), focusing only on the sequence spanning the *ARHGAP11B* exon5–exon6 boundary, where *ARHGAP11B* is sufficiently different from *ARHGAP11A* (see *Figure 5—figure supplement 1*) (*Florio et al., 2016*), and searching for hybridization with a predicted RNA melting temperature of 85°C. The LNA probe (5'-AGTCTGG TACACGCCCTTCTTTTCT-3') was synthesized and labeled with digoxigenin at the 5' and 3' ends (QIAGEN). Using the same melting temperature settings, the LNA probes for *GTF2H2C* (5'-TCA-GACGGCCTGCC-3'), *FAM182B* (5'-GCAGGGATACACGGCTAT-3') and *DHRS4L2* (5'-AGACAG TGGCGGTTGCGTGA-3') were designed accordingly to target unique sequences of the respective transcripts.

In-situ hybridization was performed on 12–µm cryosections of 13 wpc fetal human neocortex and on COS-7 cells. Prior to the hybridization step, cryosections/cells were sequentially treated with 0.2 M HCl (2 × 5 min, room temperature) and then with 6 µg/ml proteinase K in PBS, pH 7.4 (20 min, room temperature). Hybridization was performed overnight at 65°C with either 20 ng/µl of a given RNA probe or 40 nM LNA probe. TSA Plus DIG detection Kit (Perkin Elmer) was used for signal amplification, and the signal was detected immunohistochemically with mouse anti-digoxigenin HRP antibody (Perkin Elmer) and NBT/BCIP (Roche) as color substrate.

## Image acquisition

ISH images were acquired on a Zeiss Axio Scan slide scanner, and processed using ImageJ. Fluorescent images of electroporated neocortex were acquired using a Zeiss laser scanning confocal microscope 700 using a 20x objective. Quantifications were performed using Fiji.

## Acknowledgements

We are grateful to the Computer Service Facilities of the MPI-CBG and MPI-PKS and to other services and facilities of the MPI-CBG for the outstanding support provided, notably J. Helppi and his team from the Animal Facility, P. Keller and his team from the Antibody Facility, Jan Peychl and his team from the Light Microscopy Facility, and Ian Henry and his team from the Scientific Computing Facility. We thank Robert Lachmann for providing fetal human tissue, Hella Hartmann of the CRTD of the Technische Universität Dresden for help with the acquisition of the ISH images, Virag Sharma for help with CESAR, and members of the Huttner laboratory for critical discussion. We are grateful to Dr. Tomislav Maricic (MPI for Evolutionary Anthropology) for donating human, chimpanzee and bonobo genomic DNA. M.F. would like to thank Dr. Fenna Krienen (Harvard Medical School) for helpful discussion and critical reading of the manuscript. M.F. was a member of the International Max Planck Research School for Cell, Developmental and Systems Biology and a doctoral student at Technische Universität Dresden. W.B.H. was supported by grants from the Deutsche Forschungsgemeinschaft (DFG) (SFB 655, A2), the European Research Council (250197), and ERA-NET NEURON (MicroKin).

# Additional information

## Funding

| Funder | Grant reference number | Author |
|---|---|---|
| Max-Planck-Gesellschaft | | Wieland B Huttner<br>Michael Hiller |
| Deutsche Forschungsge-meinschaft | SFB655 A2 | Wieland B Huttner |
| European Research Council | 250197 | Wieland B Huttner |

The funders had no role in study design, data collection and interpretation, or the decision to submit the work for publication.

## Author contributions

Marta Florio, Conceptualization, Formal analysis, Investigation, Visualization, Methodology, Writing—review and editing; Michael Heide, Conceptualization, Formal analysis, Investigation, Visualization, Methodology, Writing—original draft, Writing—review and editing; Anneline Pinson, NOTCH2NL experiments; Holger Brandl, Bioinformatics; Mareike Albert, qPCR validation of the of the paralog-specific gene expression analysis; Sylke Winkler, quantitative PCR; Pauline Wimberger, Resources: fetal human neocortical tissue; Wieland B Huttner, Conceptualization, Supervision, Funding acquisition, Writing—original draft, Project administration, Writing—review and editing; Michael Hiller, Conceptualization, Formal analysis, Supervision, Visualization, Methodology, Writing—review and editing

## Author ORCIDs

Holger Brandl (iD) http://orcid.org/0000-0003-1911-8570
Wieland B Huttner (iD) http://orcid.org/0000-0003-4143-7201

## Ethics

Human subjects: Human fetal brain tissue (12-13 weeks post conception (wpc)) was obtained from the Klinik und Poliklinik für Frauenheilkunde und Geburtshilfe, Universitätsklinikum Carl Gustav Carus of the Technische Universität Dresden with informed written maternal consent followed by elective pregnancy termination. Research involving human tissue was approved by the Ethical Review Committee of the Universitätsklinikum Carl Gustav Carus of the Technische Universität Dresden (reference number: EK100052004). In addition, research was approved by the Institutional Review Board of the Max Planck Institute of Molecular Cell Biology and Genetics.
Animal experimentation: All animal experiments were performed in accordance with German animal welfare laws and overseen by the institutional review board (reference number TVV2015/05). C57BL/6J mice were maintained in specific pathogen-free conditions in the MPI-CBG animal facility.

## Decision letter and Author response

Decision letter https://doi.org/10.7554/eLife.32332.044
Author response https://doi.org/10.7554/eLife.32332.045

# Additional files

## Supplementary files

• Supplementary file 1. cNPC-enriched genes. This file summarizes information of the five datasets, occurrence of all cNPC-enriched genes in the five datasets and composition of the five gene sets including gene expression data.
DOI: https://doi.org/10.7554/eLife.32332.024

• Supplementary file 2. GO term analysis of cNPC-enriched genes. This file contains the output of the GO term analysis.

DOI: https://doi.org/10.7554/eLife.32332.025

• Supplementary file 3. Chromosome location of all cNPC-enriched primate-specific genes in the different primates. This file contains the chromosome location of all cNPC-enriched primate-specific genes in the 12 primate species analyzed.
DOI: https://doi.org/10.7554/eLife.32332.026

• Supplementary file 4. mRNA expression data of splice variants. This file contains mRNA expression data for the human-specific genes and their corresponding ancestral paralog for each cell type and splice variant, including non-coding transcripts.
DOI: https://doi.org/10.7554/eLife.32332.027

• Supplementary file 5. qPCR primer. This file contains the primer sequences of the qPCR for the validation of the paralog-specific gene expression analysis.
DOI: https://doi.org/10.7554/eLife.32332.028

• Supplementary file 6. Primer for genomic qPCR. This file contains the primer sequences of the genomic qPCR.
DOI: https://doi.org/10.7554/eLife.32332.029

• Supplementary file 7. Primer for ISH probes. This file contains the primer sequences used to generate the templates for the synthesis of the ISH probes.
DOI: https://doi.org/10.7554/eLife.32332.030

• Transparent reporting form
DOI: https://doi.org/10.7554/eLife.32332.031

## Major datasets

The following previously published datasets were used:

| Author(s) | Year | Dataset title | Dataset URL | Database, license, and accessibility information |
| --- | --- | --- | --- | --- |
| Fietz SA, Huttner WB, Pääbo S | 2012 | Transcriptomes of germinal zones of human and mouse fetal neocortex suggest a role of extracellular matrix in progenitor self-renewal | https://www.ncbi.nlm.nih.gov/geo/query/acc.cgi?acc=GSE38805 | Publicly available at the NCBI Gene Expression Omnibus (accession no. GSE38805) |

| | | | | |
|---|---|---|---|---|
| Miller JA, Ding SL, Sunkin SM, Smith KA, Ng L, Szafer A, Ebbert A, Riley ZL, Royall JJ, Aiona K, Arnold JM, Bennet C, Bertagnolli D, Brouner K, Butler S, Caldejon S, Carey A, Cuhaciyan C, Dalley RA, Dee N, Dolbeare TA, Facer BA, Feng D, Fliss TP, Gee G, Goldy J, Gourley L, Gregor BW, Gu G, Howard RE, Jochim JM, Kuan CL, Lau C, Lee CK, Lee F, Lemon TA, Lesnar P, McMurray B, Mastan N, Mosqueda N, Naluai-Cecchini T, Ngo NK, Nyhus J, Oldre A, Olson E, Parente J, Parker PD, Parry SE, Stevens A, Pletikos M, Reding M, Roll K, Sandman D, Sarreal M, Shapouri S, Shapovalova NV, Shen EH, Sjoquist N, Slaughterbeck CR, Smith M, Sodt AJ, Williams D, Zöllei L, Fischl B, Gerstein MB, Geschwind DH, Glass IA, Hawrylycz MJ, Hevner RF, Huang H, Jones AR, Knowles JA, Levitt P, Phillips JW, Sestan N, Wohnoutka P, Dang C, Bernard A, Hohmann JG, Lein ES | 2014 | Transcriptional landscape of the prenatal human brain | http://www.brainspan.org/lcm/search/index.html | Available at the Allen Brain Atlas |
| Florio M, Albert M, Huttner WB | 2015 | Human-specific gene ARHGAP11B promotes basal progenitor amplification and neocortex expansion | https://www.ncbi.nlm.nih.gov/geo/query/acc.cgi?acc=GSE65000 | Publicly available at the NCBI Gene Expression Omnibus (accession no. GSE65000) |
| Pollen AA, Nowakowski TJ, Chen J, Retallack H, Sandoval-Espinosa C, Nicholas CR, Shuga J, Liu SJ, Oldham MC, Diaz A, Lim DA, Leyrat AA, West JA, Kriegstein AR | 2015 | Molecular Identity of Human Outer Radial Glia Cells During Cortical Development | https://www.ncbi.nlm.nih.gov/projects/gap/cgi-bin/study.cgi?study_id=phs000989.v1.p1 | Publicly available at the NCBI Gene Expression Omnibus (Study Accession no. phs000989.v1.p1) |
| Walsh CA, Johnson MB, Wang PP | 2015 | Single-cell analysis reveals transcriptional heterogeneity of neural progenitors in human cortex | https://www.ncbi.nlm.nih.gov/geo/query/acc.cgi?acc=GSE66217 | Publicly available at the NCBI Gene Expression Omnibus (accession no. GSE66217) |

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
