## [Decision Letter]

Thank you for submitting your article "Evolution and cell-type specificity of human-specific genes preferentially expressed in progenitors of fetal neocortex" for consideration by *eLife*. Your article has been reviewed by three peer reviewers, and the evaluation has been overseen by a Reviewing Editor and Didier Stainier as the Senior Editor. The reviewers have opted to remain anonymous.

The reviewers have discussed the reviews with one another and the Reviewing Editor has drafted these comments to help you consider how to proceed. While the reviews were favorable in general, and felt that the topic was important, there were concerns about the reliability of both the computational as well as the experimental data, which dampened their enthusiasm. As you can see from the critiques below, the issues can be divided into three categories: 1] Applying more rigorous statistical analysis of the list of genes that are human- or primate-specific, to ensure that the list is both comprehensive and reliable. 2] Re-analysis of the in situ hybridization data to ensure that the probes being used are specific for the transcript(s) of interest. 3] Re-analysis of the in utero electroporation with additional controls to ensure that potential off-target effects are taken into account. Due to the number of these criticisms, *eLife* would understand if you decided to move the manuscript to another journal, to avoid lengthy revisions. But performing these revisions to the level to satisfy reviewers might be relatively straightforward for you, because these might be issues that you have previously addressed or can address in a straightforward way or with more detailed explanations of your methodology. We are interested in seeing this work published in *eLife*, and would like to put this question to you, and hear your preference for how to proceed.

Summary:

Florio et al. describe their efforts to mine existing transcriptomic data sets for genes that may have importance for human neurodevelopment. Building on their experience with the human-specific *ARHGAP11B*, they focused on genes that are either human- or primate-specific, with a strong focus on the former. They identified several such genes and characterized them by a combination of detailed expression analysis using the existing data, novel in situ hybridization approaches, as well as functional overexpression in the mouse cortex for one of the genes. They also provide analysis of the origin and evolutionary history of these genes.

This manuscript presents a laudable effort to combine the various transcriptome data sets that exist in this field and were generated by several labs. This allows to overcome limitations of the individual studies and to provide a more definitive answer to 'what are the defining genetic features of human brain development?" Even as a candidate for a 'Tools and Resources' contribution, there was a sense that the analysis was too preliminary for general conclusions.

Essential revisions:

1) Almost all of their in situ hybridization experiments are unable to distinguish between the ancestral paralogs and the genes of interest due to probe cross-reaction, and in those cases where they do, the original gene is not shown. Could this problem not be circumvented using alternative RNA detection methods such as RNAscope or using LNAs (as done for one of the genes)? Given the lack of ability to distinguish between paralogs, the claim that one can distinguish gene expression patterns is quite bold. As it stands, very little can be deduced from these data, unless gene-specific ISH can be performed.

2) The authors do not attempt any other cross-validation of the results presented in Figure 7 (which does not include all genes they describe) with qPCR, ddPCR, with or without a TaqMan strategy, or alternative method. Without this data, it is hard to accept the claims. In my opinion, there should be some independent way to validate this, especially if the authors want to draw such attention to their finding.

3) Their selection of data sets omits the prominent example published by Johnson et al., 2015. While the authors might have reasons to exclude these data, this could be a very powerful data set as it is technically more similar to the Florio et al. data set than some of the others used.

4) 3722 genes is a large proportion of the known genes, therefore any of these might be just found by coincidence. An enrichment analysis would useful to support the authors' claim that there is indeed an enrichment for cell types and functions that one would expect based on their approach. As a side note, this section might also be improved by giving more concrete references for the markers, so the reader does not have to accept this statement at face value.

5) The authors only functionally test one of the genes, which will leave readers wondering why others were not tested. What criteria did the authors use for choosing the specific genes to study molecular mechanisms of evolution?

The *NOTCH2NL* gene is a very interesting human-specific gene, as described in the analysis. However, the experiments in Figure 9 require more controls to justify the strong interpretation presented here, i.e. control that expresses the original *NOTCH2* to determine differences and similarities. While interesting, the experiments add little to the resource value of the paper and ought to be either strengthened or removed.

6) In the third paragraph of the subsection “Spatial mRNA expression analysis in fetal human neocortex of the human-specific cNPC-enriched protein-coding genes and of three selected primate- but not human-specific protein-coding genes”, the authors argue that those genes displaying the ISH expression pattern VZ,CP>SVZ (high in both VZ and CP, while lower in SVZ) still fulfill the initial screening criterion of genes highly enriched in germinal zones. Their argument is that, for these genes, the sum of mRNA levels by RNAseq in the three germinal zones is greater than in CP. This argument is flawed because, to make this comparison, RNAseq data values should be averaged proportionally to cell density, which in any case the result would never be greater than the value observed in the individual layer with highest amount.

7) In utero electroporations are frequently subject to artifact. In situ validation that the cells in the *NOTCH2NL* electroporated samples showing ectopic cycling and abventricular localization actually express *NOTCH2NL* should be presented. This is particularly important in light of the small number of abventricular cells in panel S1E.

8) The differential gene expression strategy seems reasonable, but many times p-value does not capture the complexity of the different aspects of differential expression. Including a table with details about fold change, percent of cells expressing, and p-value for each dataset would be valuable for individuals who may have additional considerations than those presented in the paper, but precludes the need to repeat all the analysis. As such, including p-values up to 0.05 or even a bit higher would be most valuable, and the authors should highlight which genes were selected from each dataset for their analysis.

9) None of the papers from which data was analyzed carefully considered the implication of number of individuals on analysis – particularly for the single cell datasets, but also for the bulk analyses. It would be helpful to summarize for at least their top candidates in how many individuals the pattern was consistent with the overall significant observation (across datasets), and in how many the enrichment was non-existent or opposite. If any of the genes do not hold up in this analysis, they should be removed from the main figure gene lists.

10) The analysis of genomic remodeling during human evolution is nice, but readers will wonder how frequently did one or other type of event take place? Given that the authors discovered not so many genes newly evolved in human or hominids, it is essential to perform similar analyses on a few more genes to provide a global view of which events of genetic evolution occurred more frequently, and even venture to speculate why some events occurred more frequently than others.

11) Figure S1 – Bottom panels for PH3 analysis at E14.5 after electroporation 1 day before: the graph and pictures show near-null presence of basal (abventricular) mitoses in control embryos, which is clearly not the case under normal circumstances, especially at this intermediate stage of cortical neurogenesis. How is this possible, even in GFP- cells?

---

## [Author Response]

Essential revisions:1) Almost all of their in situ hybridization experiments are unable to distinguish between the ancestral paralogs and the genes of interest due to probe cross-reaction, and in those cases where they do, the original gene is not shown. Could this problem not be circumvented using alternative RNA detection methods such as RNAscope or using LNAs (as done for one of the genes)? Given the lack of ability to distinguish between paralogs, the claim that one can distinguish gene expression patterns is quite bold. As it stands, very little can be deduced from these data, unless gene-specific ISH can be performed.

As requested, we have now used specific probes for those human-specific genes for which the mRNA nucleotide sequence is sufficiently different to the respective ancestral paralog to design such probes. This was the case for *ARHGAP11B, NOTCH2NL, DHRS4L2, FAM182B, GTF2H2C* and *ZNF492*. In addition, we designed probes that selectively target the ancestral paralogs, wherever possible, to allow comparison with the cellular distribution of the mRNA of the respective human-specific gene. This was the case for *ARHGAP11A* and *NOTCH2*. These new results are described in the main text and in Figure 5. For *ANKRD20A2/4* vs. *ANKRD20A1, CBWD6* vs. *CBWD1, FAM72B/C/D* vs. *FAM72A*, and *SMN2* vs. *SMN1*, the nucleotide differences are simply too small and therefore it is not possible to design specific probes that selective target the mRNAs of these genes only. Therefore, we present the results of probes that target these paralog families.

2) The authors do not attempt any other cross-validation of the results presented in Figure 7 (which does not include all genes they describe) with qPCR, ddPCR, with or without a TaqMan strategy, or alternative method. Without this data, it is hard to accept the claims. In my opinion, there should be some independent way to validate this, especially if the authors want to draw such attention to their finding.

As requested by the reviewers, we have sought to validate by qPCR the differential expression between ancestral vs. human-specific paralogs originally presented in Figure 7. We were able to design primers detecting specific expression of ancestral vs. human-specific paralog pairs in the case of *ARHGAP11A/B, GTF2H2/GTF2H2C, NOTCH2/NOTCH2NL* and *ZNF98/ZNF492*, and we performed a qPCR analysis using cDNA libraries previously prepared from pools of cNPCs and neurons, FACSisolated from fetal human neocortex (Florio et al., 2015) – thus validating our data in the same cell populations analyzed in Figure 7. We were also able to independently detect *SMN1/2* in our Kallisto data re-analysis of the Florio 2015 dataset, thus extending our previous analysis to 12 of the 14 human-specific gene duplications contained in our list.

3) Their selection of data sets omits the prominent example published by Johnson et al., 2015. While the authors might have reasons to exclude these data, this could be a very powerful data set as it is technically more similar to the Florio et al. data set than some of the others used.

We would like to thank the reviewers for this important suggestion. We have now included the Johnson et al., 2015 transcriptome dataset in our analysis and performed a comprehensive re-analysis of the combined 5 datasets. Our re-analysis, described in the updated Figures 1 and 2, yielded 3,458 human cNPC-enriched protein-coding genes, 50 of which are primate-specific. While most of the genes reported in the original submission are still included in our new gene set, this analysis allowed us to identify two additional human-specific genes (*NBPF10, NBPF14*) and four additional primate-specific genes (*GLUD2, MT1M, SLFN13, ZNF730*).

4) 3722 genes is a large proportion of the known genes, therefore any of these might be just found by coincidence. An enrichment analysis would useful to support the authors' claim that there is indeed an enrichment for cell types and functions that one would expect based on their approach. As a side note, this section might also be improved by giving more concrete references for the markers, so the reader does not have to accept this statement at face value.

As requested by the reviewers, we have performed a GO term and pathway enrichment analysis using the final set of the 3,458 human cNPC-enriched genes as input. Despite its large size, this gene set is highly enriched in terms and pathways related to cell cycle, especially mitotic cell division. Top-enriched GO terms in the categories of biological process and cellular component are shown in the new Figure 1F, and the output of the analysis is shown in Supplementary file 2.

Further, addressing the reviewers’ concern, we have added references to original papers and reviews discussing the significance of the cNPC marker genes contained in our gene set, and highlighted this in the Results section.

5) The authors only functionally test one of the genes, which will leave readers wondering why others were not tested. What criteria did the authors use for choosing the specific genes to study molecular mechanisms of evolution?The NOTCH2NL gene is a very interesting human-specific gene, as described in the analysis. However, the experiments in Figure 9 require more controls to justify the strong interpretation presented here, i.e. control that expresses the original NOTCH2 to determine differences and similarities. While interesting, the experiments add little to the resource value of the paper and ought to be either strengthened or removed.

We selected *NOTCH2NL* in light of the pivotal role played by Notch signaling in cNPC proliferation and lineage progression, which made *NOTCH2NL* stand out as a remarkably promising candidate to exert a role in corticogenesis. This has now been clarified better in the main text.

In line with the request of the reviewers, we have now performed several additional experiments to strengthen the validity of our *NOTCH2NL* results. We believe that the data provided now conclusively show that expression of the human-specific gene *NOTCH2NL* in mouse embryonic neocortex is sufficient to expand the pool of basal progenitors. Specifically, we have performed immunofluorescence for PCNA as an additional marker of cycling cells, thus confirming our previous data obtained by Ki67 immunostaining. Furthermore, by immunofluorescence for Tbr2 and Sox2 we could show that the increase in cycling basal progenitors is due to an increase in cycling bIPs (Tbr2-positive) rather than in cycling bRG (Sox2-positive). Finally, by determining the rate of cell cycle exit in the SVZ by Ki67 and BrdU immunofluorescence staining, we observed a decrease in cell cycle exit, thus confirming an increase in cycling cells. These new data are shown in the new main Figure 9 and are described in the last section of Results.

6) In the third paragraph of the subsection “Spatial mRNA expression analysis in fetal human neocortex of the human-specific cNPC-enriched protein-coding genes and of three selected primate- but not human-specific protein-coding genes”, the authors argue that those genes displaying the ISH expression pattern VZ,CP>SVZ (high in both VZ and CP, while lower in SVZ) still fulfill the initial screening criterion of genes highly enriched in germinal zones. Their argument is that, for these genes, the sum of mRNA levels by RNAseq in the three germinal zones is greater than in CP. This argument is flawed because, to make this comparison, RNAseq data values should be averaged proportionally to cell density, which in any case the result would never be greater than the value observed in the individual layer with highest amount.

In line with the reviewers' comment, we have deleted this paragraph.

7) In utero electroporations are frequently subject to artifact. In situ validation that the cells in the NOTCH2NL electroporated samples showing ectopic cycling and abventricular localization actually express NOTCH2NL should be presented. This is particularly important in light of the small number of abventricular cells in panel S1E.

As requested by the reviewer, we have electroporated an HA-tagged version of *NOTCH2NL*. Author response image 1 shows high magnification images of electroporated cells in the SVZ of two independent samples. *NOTCH2NL* expression was detected by an anti-HA antibody. Abventricular cells are clearly HA-positive and show a strong correlation with GFP-expressing cells, indicating that abventricular cells express NOTCH2NL after electroporation.

8) The differential gene expression strategy seems reasonable, but many times p-value does not capture the complexity of the different aspects of differential expression. Including a table with details about fold change, percent of cells expressing, and p-value for each dataset would be valuable for individuals who may have additional considerations than those presented in the paper, but precludes the need to repeat all the analysis. As such, including p-values up to 0.05 or even a bit higher would be most valuable, and the authors should highlight which genes were selected from each dataset for their analysis.

As requested by the reviewers, we have now expanded our Supplementary file 1 to show the expression levels, differential expression values and/or correlation scores of all genes in all five gene sets. Comparison of these five lists of genes with the list of the 3,458 cNPC-enriched genes allows the identification of the genes selected from each gene set.

9) None of the papers from which data was analyzed carefully considered the implication of number of individuals on analysis – particularly for the single cell datasets, but also for the bulk analyses. It would be helpful to summarize for at least their top candidates in how many individuals the pattern was consistent with the overall significant observation (across datasets), and in how many the enrichment was non-existent or opposite. If any of the genes do not hold up in this analysis, they should be removed from the main figure gene lists.

We would like thank the reviewers for raising this important point. We have added two supplementary figures (Figure 7—figure supplement 3 and 4) showing gene expression profiles of all the cNPC-enriched human-specific genes, in all datasets analyzed, across all samples and individuals. This analysis showed that virtually all genes analyzed were expressed in all individual specimens studied (with the exception of *NBPF10/14* in the Florio dataset, where these genes were not detected altogether; Figure 7—figure supplement 3). Moreover, since the Fietz dataset sampled six different fetal samples from four distinct gestational ages (13-16 wpc), we could provide information on the temporal progression of expression of these genes during corticogenesis (Figure 7—figure supplement 3).

10) The analysis of genomic remodeling during human evolution is nice, but readers will wonder how frequently did one or other type of event take place? Given that the authors discovered not so many genes newly evolved in human or hominids, it is essential to perform similar analyses on a few more genes to provide a global view of which events of genetic evolution occurred more frequently, and even venture to speculate why some events occurred more frequently than others.

We have repeated our previous analysis on the evolutionary origin of human-specific genes, extending it to the new list of human-specific cNPC-enriched genes resulting from the inclusion of the Johnson et al., 2015 dataset. As described in the Results and presented in Figure 4, this analysis shows that 13 of the 15 human-specific genes identified here evolved by entire or partial gene duplication. The origin of the other two genes involved exon duplication, leading to a chimeric zinc finger gene, and the removal of a premature stop codon. Therefore, while gene duplication is the main “*genomic remodeling*” evolutionary drive in our dataset, we were able to observe and describe at least two interesting exceptions.

11) Figure S1 – Bottom panels for PH3 analysis at E14.5 after electroporation 1 day before: the graph and pictures show near-null presence of basal (abventricular) mitoses in control embryos, which is clearly not the case under normal circumstances, especially at this intermediate stage of cortical neurogenesis. How is this possible, even in GFP- cells?

We agree with this criticism of the reviewers. The image shown in Figure S1C left was a single optical section and thus was not really representative. We have replaced the images with more representative ones, which are now shown in Figure 9J.